# Phylogenomic analyses of all species of swordtail fishes (genus *Xiphophorus*) show that hybridization preceded speciation

Kang Du [1], Juliana Morena Bonita Ricci[1], Yuan Lu[1], Mateo Garcia-Olazabal[1], Ronald B. Walter[1], Wesley C. Warren[2], Tristram O. Dodge [3], Molly Schumer[3], Hyun Park[4], Axel Meyer [5] ✉ & Manfred Schartl [1,6,7] ✉

Hybridization has been recognized to play important roles in evolution, however studies of the genetic consequence are still lagging behind in vertebrates due to the lack of appropriate experimental systems. Fish of the genus *Xiphophorus* are proposed to have evolved with multiple ancient and ongoing hybridization events. They have served as an informative research model in evolutionary biology and in biomedical research on human disease for more than a century. Here, we provide the complete genomic resource including annotations for all described 26 *Xiphophorus* species and three undescribed taxa and resolve all uncertain phylogenetic relationships. We investigate the molecular evolution of genes related to cancers such as melanoma and for the genetic control of puberty timing, focusing on genes that are predicted to be involved in pre-and postzygotic isolation and thus affect hybridization. We discovered dramatic size-variation of some gene families. These persisted despite reticulate evolution, rapid speciation and short divergence time. Finally, we clarify the hybridization history in the entire genus settling disputed hybridization history of two Southern swordtails. Our comparative genomic analyses revealed hybridization ancestries that are manifested in the mosaic fused genomes and show that hybridization often preceded speciation.

The debate in evolutionary biology on the role and importance of hybridization and postzygotic hybrid sterility genes in the process of speciation remains unsettled. For decades, hybridization was thought to be rare in animals and perhaps particularly so in vertebrates[1]. However, this view was overturned by an increasing number of genomic studies over the last two decades. With this growing appreciation for the ubiquity of hybridization, researchers have become interested in the evolutionary consequences of hybridization, particularly for adaptation and as a mechanism of speciation[1–5]. While examples of reticulate evolution[6,7] are increasing, evidence for hybrid speciation in vertebrates is still rare.

*Xiphophorus* is a genus of Central American freshwater fishes[8]. To date, 26 species have been described that occur in various freshwater habitats in the Atlantic drainages of Mesoamerica, from Northern Mexico to Guatemala. Historically, they have been classically divided into four groups according to their geographic distribution: the

[1]The Xiphophorus Genetic Stock Center, Texas State University, San Marcos, Texas, TX, USA. [2]Department of Animal Sciences, Department of Surgery, Institute for Data Science and Informatics, University of Missouri, Bond Life Sciences Center, Columbia, MI, USA. [3]Department of Biology & Howard Hughes Medical Institute, Stanford University, Stanford, CA, USA. [4]Division of Biotechnology, College of Life Science and Biotechnology, Korea University, Seoul, Republic of Korea. [5]Department of Biology, University of Konstanz, Konstanz, Germany. [6]Developmental Biochemistry, Biocenter, University of Wuerzburg, Am Hubland, Wuerzburg, Germany. [7]Research Department for Limnology, University of Innsbruck, Mondsee, Austria. ✉e-mail: axel.meyer@uni-konstanz.de; phch1@biozentrum.uni-wuerzburg.de

Northern and Southern platyfishes and the Northern and Southern swordtails. Their striking morphological variation and the possibility to experimentally produce inter-species hybrids[9,10] render *Xiphophorus* an ideal vertebrate model system to address questions related to the role of hybridization in phenotypic evolution and speciation. In previous work, for two species, an origin from a hybridization event was proposed[11–14], and a transcriptome-based survey revealed evidence for reticulate evolution[15]. Moreover, there are several known ancient[16] and contemporary hybrid zones in this group[17,18]. Although hybridization, especially within *Xiphophorus*, appears to be more frequent than previously thought, the evolutionary impact of hybridization will be shaped by the extent to which hybrids between lineages survive and reproduce and thereby introduce novel genetic material into different lineages horizontally rather than vertically.

In nature, both pre- and postzygotic isolation mechanisms impact the formation and persistence of hybrids. Prezygotic isolation can be mediated by species-specific differences in courtship and mating behavior, among other mechanisms[19,20]. *Xiphophorus* fish, commonly called swordtails and platyfishes, have been intensively studied in part because of their dramatic sexually selected ornaments[21–23]. The most well studied sexually selected traits in swordtails – first noted by Darwin – is a colorful extension of the ventral rays of the caudal fin referred to as the "sword"[24]. This trait is found in all species of Southern swordtails and most Northern swordtails but is absent in platyfishes[8]. The sword ornament is used in courtship displays and is highly attractive to females in many species, even those whose males do not have swords[25,26] and serves as a reproductive barrier in species that have lost the sword[27]. The "preexisting bias" hypothesis postulates that the female preference is ancestral and that it thereby facilitated the later evolution of the males' trait[28]. Knowledge about a mono- or polyphyletic origin of the sword in combination with a precise species tree is required to validate this hypothesis[14]. Candidate "sword development genes" responsible for an overgrowth of fin rays have been proposed for two loci on chromosome 13[29,30], but their true involvement in the production of this spectacular trait remains unproven.

Differences in life-history traits involved with reproduction may be barriers to hybridization. In several species of *Xiphophorus,* the onset of male sexual maturity is genetically determined by so-called puberty loci (*P*) on the sex chromosomes[31]. In two species, it was shown that the *P* loci harbor various wildtype and defective mutant alleles encoding melanocortin 4 receptors, Mc4r[32]. This gene is known to be involved in metabolic regulation, obesity, and the onset of puberty in mammals[33]. Although a molecular mechanism for the action of dominant-negative versions of Mc4r in regulating the onset of the reproductive period and associated traits such as adult size, courtship, and dominance behaviors, has been revealed[34–36], the phylogenetic distribution of this system and its evolution are not known yet.

The evolutionary outcomes of hybridization also depend crucially on genetic interactions that occur when the genomes of two divergent lineages are combined. Postzygotic isolation in *Xiphophorus* became a textbook example of a "speciation gene"[37–39], or a genetic interaction that generates reproductive barriers between lineages. This emerged from studies on the spontaneous development of melanoma among select *Xiphophorus* interspecies hybrids[40,41]. Several laboratory hybrid crosses, as well as natural hybrid populations of *Xiphophorus* species[42], develop malignant pigment cell lesions and serve as biomedical models for understanding the genetic etiology of the human disease in translational studies[43,44]. A mutant copy of an epidermal growth factor receptor (*egfrb*) allele, *xmrk*, has been detected in *X. maculatus*[45,46] and acts as a melanoma-inducing oncogene. Normally, in *X. maculatus,* the *xmrk* oncogene is kept in check by an epistatic trans-acting regulatory locus *R/Diff*, while in hybrids crossing conditioned-elimination of *R/Diff* leading to *xmrk* dysregulation causes cancer. Previous studies indicated that *xmrk* is absent in some *Xiphophorus* species and that not all interspecies crosses result in melanoma-bearing hybrids[47,48]. The

evolutionary relevance of the *R/Diff-xmrk* oncogene/tumor suppressor gene situation in some but not all species, and questions about a monophyletic vs convergent evolution of this trait are currently unknown.

To formally test evolutionary hypotheses for the occurrence and transmission of these and many other important traits in *Xiphophorus* species requires comprehensive and high-quality genomic resources and an accurate phylogeny. Reference genomes for several representative species of the genus have been generated, but for most of the species, annotated genome assemblies are missing. Several phylogenies based on mitochondrial sequences and/or partial nuclear genomic sequences have been forwarded[12–15] but some species' placements remained uncertain, also due to suspected ancient gene flow or hybrid origins of the species. We sequenced, assembled, and annotated the genomes of all known *Xiphophorus* species as well as three undescribed taxa of the genus *Xiphophorus* for which genomes had not been determined previously. From the complete genomic dataset for this genus, we generated both the mitochondrial and nuclear phylogenies of all *Xiphophorus* species and, based on that, studied the evolution of the genomes, selected genes, and gene families. We find evidence of extensive hybridization during the evolution of *Xiphophorus* in both current and ancestral lineages.

## Results
### Genome assembly and annotation
Five chromosome-level assembled genomes (*X. maculatus, X. hellerii, X. couchianus, X. birchmanni, X. malinche*) from the 26 so far described species of the genus *Xiphophorus* have been published[42,49]. Here, we add twenty-eight additional genomes (one on chromosome level) to these existing genomic resources, which were sequenced using Illumina, 10X, PacBio, and/or Hi-C techniques in this study (Supplementary Data 1). This now provides a complete genome resource of all *Xiphophorus* species, including all previously identified species (https://www.ncbi.nlm.nih.gov/Taxonomy/Browser/wwwtax.cgi?id=8082), as well as from three undescribed taxa, *X. sp* I, *X. sp* II, *X. sp* III, two new strains of *X. maculatus*, and a reference genome for the species *Priapella lacandonae* as outgroup.

The resulting assemblies range from ~636 Mb to 720 Mb (close to the estimated genome size[50]), with a scaffold N50 ranging from ~2 Mb to 32 Mb. BUSCO analyses demonstrate that each assembly covers ~92–97% of the 4584 well-conserved Actinopterygii genes (Supplementary Data 2). Using RepeatModeler and RepeatMasker[51], we found ~30–35% of the content of the *Xiphophorus* assemblies is made up of transposable elements (TEs) (Supplementary Data 3). When aligning each assembly to *X. maculatus* (GCA_002775205.2), the coverage is ~84–92% (Supplementary Data 4). Average divergence between genomes within the genus *Xiphophorus* was all lower than ~2.5% (Supplementary Data 5). However, as expected from both hybridization and variation in constraint and the strength of background selection[52,53], sequence divergence varies along the genome. Each chromosome has regions of higher and lower sequence conservation. Alternating regions of high or low divergence on a chromosome are observed in a consistent pattern across the whole genus (Supplementary Figs. S1, S2).

Using a custom annotation pipeline[54], we annotated ~22–25k protein-coding genes (PCGs) in each assembly, improving the BUSCO assessment of completeness by ~1% (for BUSCO values of all species, see Supplementary Data 6). Based on the similarity of protein sequences, these PCGs were clustered into 26,982 gene families (Supplementary Fig. S3). Out of these gene families, we find evidence that 353 families underwent a significant change in the size of the family during the evolution of the genus *Xiphophorus*. These include olfactory receptors, odorant receptors, retinol dehydrogenase 12, and melanocortin 4 receptors (*mc4r*) (Supplementary Figs. S4, S5 and Supplementary Data 7).

Those genomes, which are assembled at chromosome level (*X. birchmanni, X. malinche, X. couchianus, X. maculatus, X. hellerii,* and *P. lacandonae*), show, in general, a high conservation of synteny with the exception of the two platyfish species, *X. maculatus* and *X. couchianus*. They have several translocations and inversions compared to the karyotypes of the Northern and Southern swordtails (Supplementary Fig. S6).

## Gene evolution

The availability of the genomes of all swordtails and platyfishes, allowed us to determine the species-specific presence or absence and origin of genes that are important for *Xiphophorus* as a model for speciation and human diseases.

***xmrk* melanoma oncogene.** The *xmrk* gene was originally found in the Southern platyfish, *X. maculatus*. From the genome sequences (assemblies and primary reads) followed by PCR confirmation, we were able to retrieve *xmrk* sequences in a total of nine platyfish and Northern swordtail species but not in any of the Southern swordtails (Supplementary Figs. S7A, S8). This suggests that the gene duplication generating *xmrk* has occurred at the base of the platyfish and Northern swordtail clades. Due to the high degree of conservation of *xmrk* and *egfrb* (its proto-oncogenic precursor) coding sequences, several nodes of the gene tree are only poorly supported (Supplementary Fig. S7B). Despite this, a monophyletic origin of *xmrk* is visible.

**Sword candidate genes.** The search for genes involved in the evolution of the sword resulted in several suggested developmental mechanisms and candidate genes[55–59]. From genetic mapping approaches combined with spatial and temporal patterns of fin growth and differentiation genes, a promising sword candidate gene is *kcnh8*[30]. It is present in *Xiphophorus*, as in other non-polyploid teleosts, as a single copy gene. The gene tree largely follows the species tree, but no amino-acid changes were detected that correlated with the absence or presence of a sword. The same situation was found for another sword candidate gene, *sp8*, postulated in an independent study[29].

**Defective copies of *mc4r*.** The onset of reproductive maturity in males of two species (*X. nigrensis*, and *X. multilineatus*) is regulated by the number of mutant *mc4r* copies clustered on the Y chromosome (historically referred to as the *P*-locus[32]). These copies are defective for transmitting the intracellular signal from *mc4r* due to changes in the carboxy-terminal amino acid sequence. From the genome assemblies, we were able to identify multiple mutant copies, which are predicted to be defective, also in *X. maculatus, X. xiphidium, X. birchmanni, X. malinche, X. evelynae* and *X. sp* III, but not in any of the Southern swordtails. This may suggest an origin of the *P*-locus system at the base of the Northern swordtail and platyfish clades, similar to the situation of the gene duplication that produced *xmrk*. Because genome assemblies can be compromised by sequencing errors (in particular short-read genomes), further analyses are necessary to confirm the absence of such defective copies and to reconstruct the evolutionary history of the different mutant alleles and copy numbers variation between and within species.

## Discordance of mitochondrial and nuclear phylogenetics

Discordance in phylogenetic trees constructed from partial mitochondrial and nuclear sequence data in *Xiphophorus* has been previously reported in several studies that date back up to 30 years[12,14,15] – a finding that is now not uncommon in many taxa[60–62]. Such discordance can indicate a history of interspecific gene flow, such as hybridization, or can be generated by processes such as incomplete lineage sorting. With our complete genome resources, including both mitochondrial and nuclear sequences, we are now able to revisit this question at a genome-wide scale. For the mitochondrial phylogenetic tree, sequences of all 37 mitochondrial genes were aligned across the species independently and then concatenated for a maximum-likelihood estimation using RAxML. The tree was also reconstructed using Bayesian approaches as an alternative method, which generated a nearly identical topology (Supplementary Fig. S9). For the nuclear sequences, we reconstructed a phylogenomic tree based on a ~342 Mb long whole genome alignment (WGA) with stringent criteria (see section "Methods") using RAxML under the GTR + Gamma model. All nodes of the tree were fully supported, as indicated by the 1000-bootstrap replications. According to mitochondrial and nuclear phylogenomics, the genus *Xiphophorus* is divided into three monophyletic groups: platyfishes (unifying the previously distinguished Northern and Southern platyfishes), a Northern swordtail, and a Southern swordtail clade, respectively. However, the platyfishes were grouped with Southern swordtails in the mitochondrial tree but with Northern swordtails according to the nuclear phylogeny (Fig. 1). This incongruence also emerged in trees in previous studies but was not further discussed[12,14,15].

Our result also confirmed previous reports of the conflicting placement of two southern swordtail species, *X. clemenciae*[14] and *X. monticolus*[12,15]. As expected from morphological traits, in the nuclear tree, they are placed in the southern swordtail clade, but in the mitochondrial tree, they are nested within the platyfish lineage.

Based on protein-coding genes (PCG) for reconstructing the nuclear phylogeny, we built maximum-likelihood trees using 3259 one-to-one orthologous genes with their concatenated protein sequences, coding sequences, and fourfold degenerate sites (4DTV). The resulting phylogenetic estimates were almost identical to the WGA tree (Supplementary Fig. S10), except *X. xiphidium* and *X. alvarezi* were placed in slightly different positions within their respective clade.

Discordance between PCG and WGA trees indicates incongruent phylogeny across genomic loci. We investigated this further using coalescent methods to build a nuclear phylogeny. To restrict sequence sampling to recombination-free loci, we randomly sampled 19,111 WGA blocks that are 20 kb away from each other and selected a one-kb window from each block to build the individual trees. Alignment of these trees revealed high phylogeny discordance across the genome (Fig. 2A). After removing trees with poor supporting value, we built a coalescent phylogeny using weighted ASTRAL[63,64]. Notably, in this analysis, *X. clemenciae, X. monticolus,* and *X. mixei* were placed outside of the Southern swordtails and basal to the Northern swordtails and platyfishes as a separate clade (Fig. 2B). This placement was also suggested by the analysis with CASTER, a coalescence-aware alignment-based species tree estimator[64] (Supplementary Fig. S11). When using conserved non-coding regions (CNEs) as the analysis material, the incongruent placement of this clade between coalescent- and concatenated-based phylogeny remains (Supplementary Fig. S12). This incongruence (Fig. 2B, C), together with the low concordance factor across tree nodes (Supplementary Fig. S13), may indicate a high portion of admixture in the ancestor genome of the *X. clemenciae, X. monticolus* and *X. mixei* clade. To distinguish hybridization from ILS needed further analyses (see below section "Extensive reticulate hybridization during the evolution of the genus *Xiphophorus*").

Overall, all phylogenetic trees of the whole genus *Xiphophorus* and the usage of information from whole genome sequencing are largely in accordance with placements of the species from earlier morphology-based and partial sequence-based trees, except for *X. continens*. Two previous studies found a close association of *X. continens* with *X. pygmaeus*[13,15], while all our trees place *X. continens* as the sister taxon to *X. montezumae*. Our result is consistent with another recent study on Northern swordtails[65]. Whole genome re-sequencing of the material used in the conflicting studies revealed misidentification of the material in these studies[13,15] (Supplementary Fig. S14). The analysis presented here also strengthens the position of the Northern platyfishes species as a crown group within the Southern platyfishes.

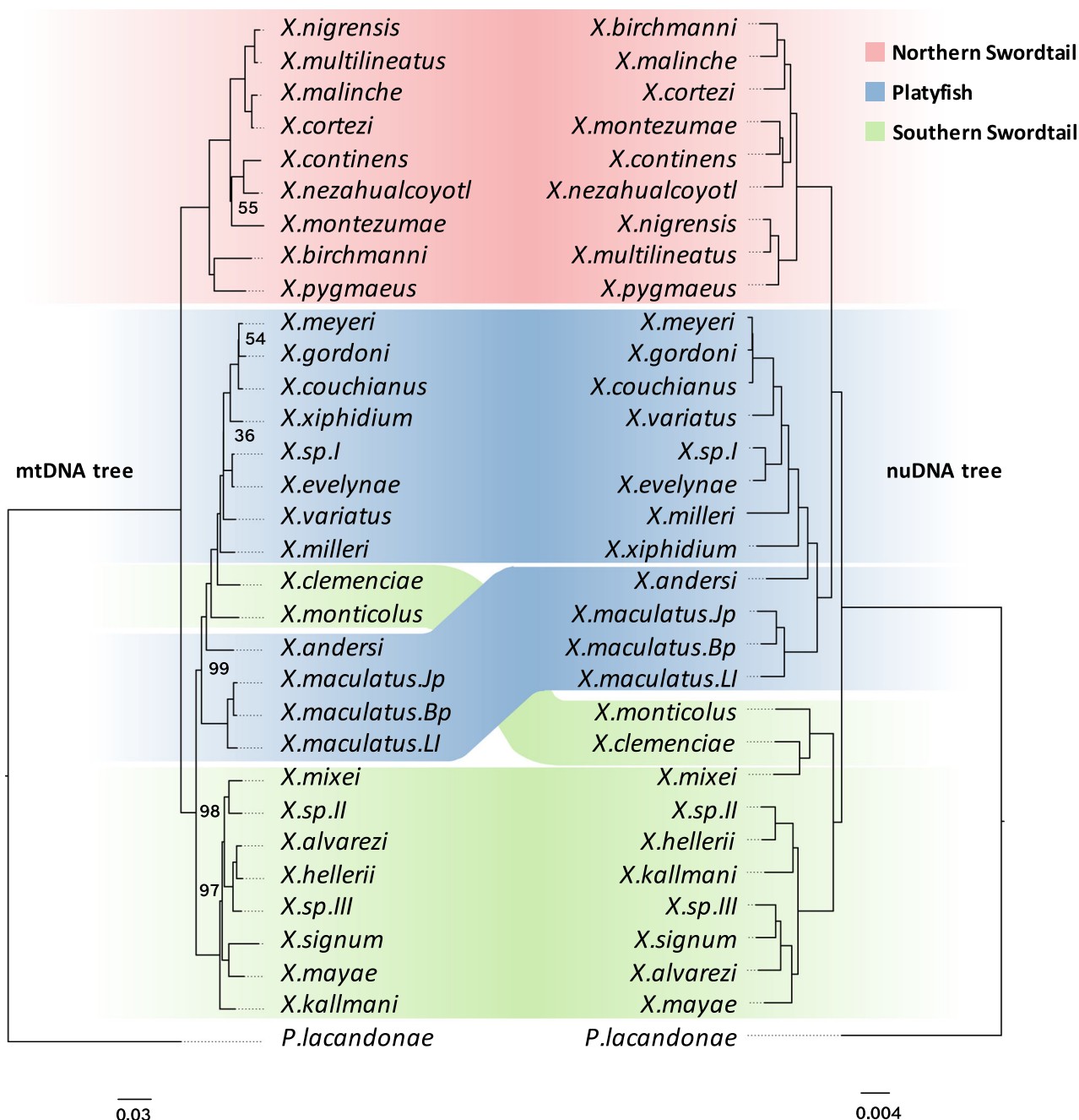

**Fig. 1 | Phylogenetic trees constructed using a maximum-likelihood method for the mitochondrial genome (~15 kb, left) and nuclear genome sequences (~342 Mb, right).** Numbers on the nodes represent bootstrap support values; nodes without numbers means that they are 100% supported.

Consequently, a distinction between both groups[8] is no longer justified and we refer to the whole clade as "platyfishes".

**Extensive reticulate hybridization during the evolution of the genus *Xiphophorus***

The presence of "platyfish-mitochondria" associated sequences in two Southern swordtail species, *X. clemenciae* and *X. monticolus*, implies that these species have experienced hybridization with the platyfish lineage during their evolution. This is in accordance with earlier data from partial mitochondrial sequences and some nuclear loci[12,14]. In addition, these Southern swordtail species (including *X. mixei*) were placed outside of the Southern swordtails and basal to the Northern swordtails and platyfishes by coalescent-based phylogeny, indicating a large non-Southern-swordtail-originated portion in their ancestor's genome.

Phylogenetic discordance can be generated by hybridization but also by other factors such as base compositional biases or rate differences, or the coalescent process via incomplete lineage sorting (ILS) that make it difficult to reconstruct the real phylogenetic relationships. It is possible to distinguish the effect of hybridization from ILS by implementing $f_4$-ratio statistics using the program Dsuite with the "*f*-branch" method and WGA data[66]. This revealed the nuclear genome of the ancestor of *X. clemenciae* and *X. monticolus* contained more than 10% admixture proportions derived from the ancestral lineage of platyfishes and Northern swordtails (Fig. 3A), consistent with a hybrid origin of *X. clemenciae* and *X. monticolus*. The fact that their maternally inherited mtDNA genome is of the platyfish type suggests that it was a female platyfish that hybridized with a male Southern swordtail. Our results also show that this hybridization must have occurred before the split of *X. clemenciae*, *X. mixei* and *X. monticolus*. This was confirmed by

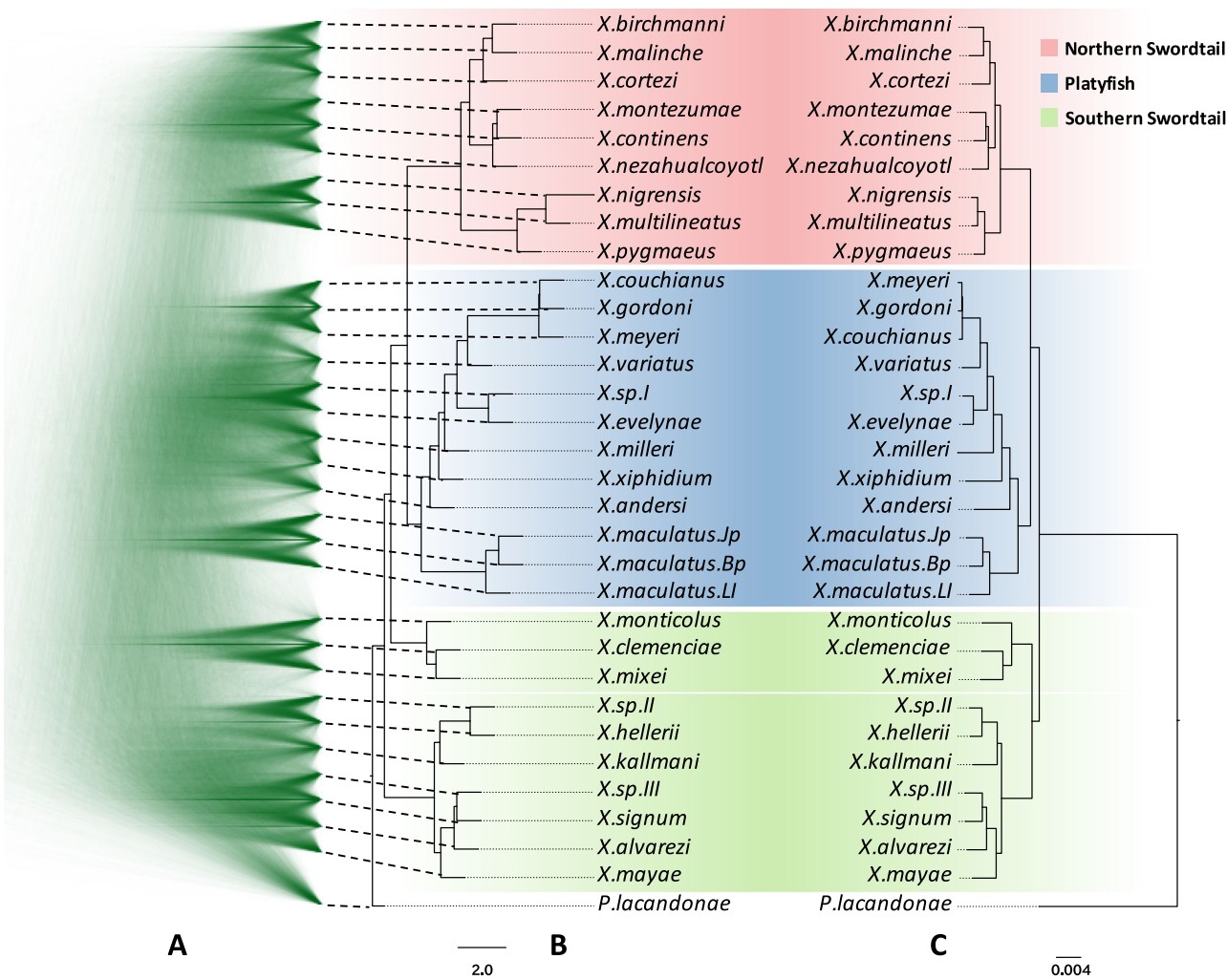

**Fig. 2 | Phylogenetic trees showing incongruent topology from coalescent- and concatenated-based methods. A** Alignment of 10,000 trees constructed from recombination-free genetic loci showing the phylogenetic discordance across the genome. **B** A coalescent-based phylogeny constructed with weightedASTRAL. **C** A concatenated-based phylogenetic tree constructed by using the maximum-likelihood method.

SNaQ in PhyloNet, a phylogenetic network analysis (Fig. 3B). A possible hybrid origin of *X. mixei* was not previously considered since mitochondrial haplotypes from these species are not discordant with the species tree. A further hybridization event was also detected between *X. mixei* and *X. monticolus* (Fig. 3A). The considerable amount of sequence originating from outside of the Southern swordtail clade may explain the more basal position of the *X. clemenciae*/*X. monticulus*/*X. mixei* subclade in the coalescent tree (Fig. 2B).

Apart from the hybrid ancestry of the *X. clemenciae, X. mixei* and *X. monticolus* subclade our analyses uncovered several other hybridization events in the genus *Xiphophorus*. A strong signal was detected in the ancestral branch leading to *X. gordoni, X. meyeri, X. couchianus, X. variatus, X. evelynae,* and *X. sp* I with *X. xiphidium* (Fig. 3A, C). Other admixture events occured between the *X. maculatus* Bp and LI populations, and between *X. signum* and *X. mayae*. In agreement with a recent phylogenomic study[65], we observe gene flow between *X. cortezi* and several other Northern swordtail species and between *X. continens* and *X. nezahualcoyotl*. Finally, we detect substantial gene flow between *X. mayae* and *X. kallmani, X. hellerii,* and *X. sp* II.

**Admixture signals across chromosomes**
We used Dsuite to explore the distribution of introgressed loci across chromosomes for species with a hybridization history. As a control, we

also included *X. alvarezi* (Fig. 4A), which did not show any signals of hybridization (Fig. 3). Our results indicate different distributions of hybridization-derived regions on different chromosomes for different hybridization events (Fig. 4). For *X. signum* (Fig. 4I), hybridization-derived regions are clustered in the middle of Chromosome 2, 15 and 18, while for other species, such regions are spread over the entire genome (Fig. 4). Conspicuously, a common admixture pattern is shared within the clade of *X. clemenciae, X. monticolus* and *X. mixei* (Fig. 4B–D) and among *X. couchianus, X. variatus, X. evelynae* and *X. sp* I (Fig. 4E–H), respectively. Local differences were found when the pattern of *X. couchianus* and *X. variatus* (Fig. 4E, F) was compared to that of *X. evelynae* and *X. sp* I (Fig. 4G, H), potentially reflecting the influence of the secondary hybridization with *X. xiphidium* (Fig. 3).

Hybridization-derived genomic regions shared by multiple species provide an opportunity to investigate the evolutionary fate of those regions following hybridization. We identified such regions in *X. clemenciae* using the AU test with WGA data. Among all the WGA regions that passed AU test (AU *P*-value > 0.95, see section "Methods"), 49.2% (95% CI: 49.0–49.3%) of them yielded the topology placing *X. clemenciae* and *X. maculatus* together (Topology 2, Supplementary Fig. S15A, B), significantly more than those placing *X. hellerii* and *X. maculatus* together (3.1%, 95% CI: 3.1–3.2%; Topology 3, Supplementary Fig. S15A, B). The estimation of the admixed partition is higher than the $f_4$ ratio test, however, it gives qualitative support for *X.*

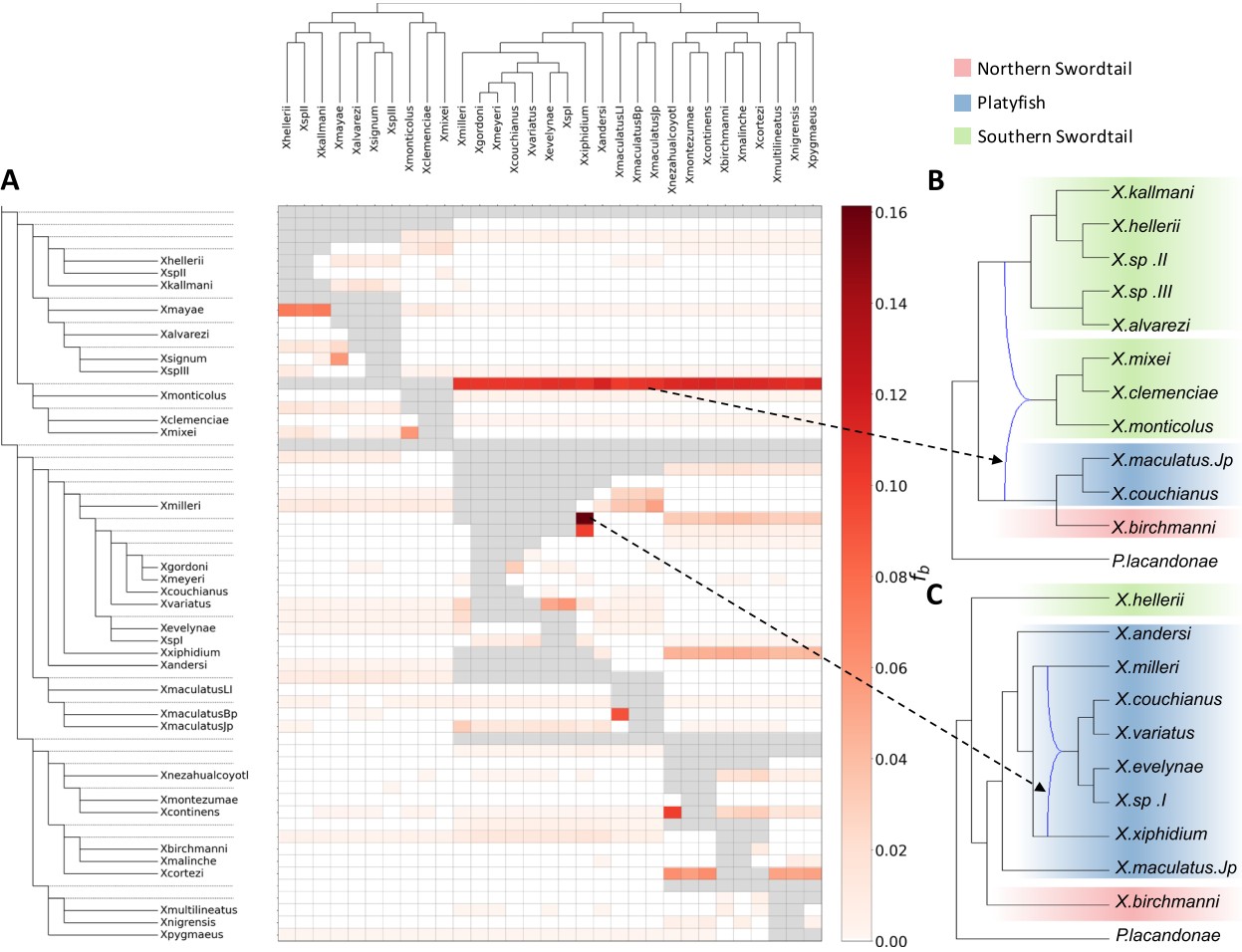

**Fig. 3 | Extensive reticulate hybridization during evolution of the genus *Xiphophorus*.** **A** Heatplot showing the estimated admixture fraction between two species on the *x*- and *y*-axis. Species are listed on the *x*- and *y*-axis in the species tree. The values in the matrix ($f_b$) were calculated using Dsuite with the *f*-branch statistic, referring to excess allele sharing between the branch identified on the expanded tree on the y-axis (relative to its sister branch) and the species identified on the x-axis, indicating the hybrid origin of these alleles. **B** Reticulate evolution of the *X. clemenciae*, *X. mixei* and *X. monticolus* clade estimated using SNaQ with *h* = 1. **C** Reticulate evolution of lineage *X. xiphidium* estimated using SNaQ with *h* = 1.

*clemenciae* being a hybrid lineage and also allows us to identify hybridization-derived regions. Those hybridization-derived regions show a lower sequence divergence (k2p value) than the others when comparing the DNA sequence between *X. clemenciae* and *X. monticolus* (Supplementary Fig. S15C). Using the same method with protein-coding gene (PCG) alignments, we also identified significantly more PCGs supporting Topology 2 than PCGs supporting Topology 3 (Supplementary Fig. S15D). Functional enrichment analysis using shinyGO[67] suggested no function is over-represented in these PCGs. Pairwise sequence comparison between *X. clemenciae* and *X. monticolus* revealed lower synonymous substitution rates (dS) and higher ratios of nonsynonymous substitution to synonymous substitution (dN/dS) for these PCGs (Supplementary Fig. S15E, F).

## Discussion

*Xiphophorus* has been used as a model to study questions from basic processes in evolutionary biology to biomedicine and human disease for over a century. In this study, we have sequenced, assembled, and annotated genomes for 19 *Xiphophorus* species and two new *X. maculatus* strains, thereby generating complete genomic resources for the whole genus *Xiphophorus*. Together with five earlier reported genomes[29,49], we provide new insights on micro- and macroevolutionary processes within the genus, generate a whole-genome based phylogeny for all species, characterize the history of hybridization and investigate the patterns of hybridization-derived regions along the genome.

## Gene evolution

Combining our robust species tree with the gene trees of *xmrk* and *egfrb* supports a single origin of *xmrk* at the base of the Northern swordtail and platyfish clades. However, many species in the Northern swordtail and platyfish clades do not have *xmrk*, suggesting that it has been lost several times independently.

The melanoma inducing action of *xmrk* is controlled by a co-evolved unlinked tumor suppressor locus on chromosome 5 for which several candidate genes have been suggested[42,68,69]. Phylogenomic studies on congruence of allelic differences in those genes with the presence or absence of *xmrk* may help to determine about the most likely gene candidate and to understand its mode of action.

The sword candidate gene *kcnh8* encodes a potassium channel. Potassium channels are found frequently to be mutated in other fish that show overgrowth of fins[70–74]. While in those species, the respective gene is a duplicate from the teleost-specific whole genome duplication or a polyploidization event that obviously underwent sub-functionalization for regulating fin growth, we find only one copy of *kcnh8* in all *Xiphophorus* species and in *Priapella*. Because of the pleiotropic function of this gene (e.g., in neuronal cells), a structural change like those observed in other fin mutant channels would also compromise *kcnh8* in other cells. Thus, it may be more likely that a regulatory change is the basis of sword formation.

Another conclusion comes from the complete phylogeny of the genus that confirms an earlier study[14] arguing the sword has evolved

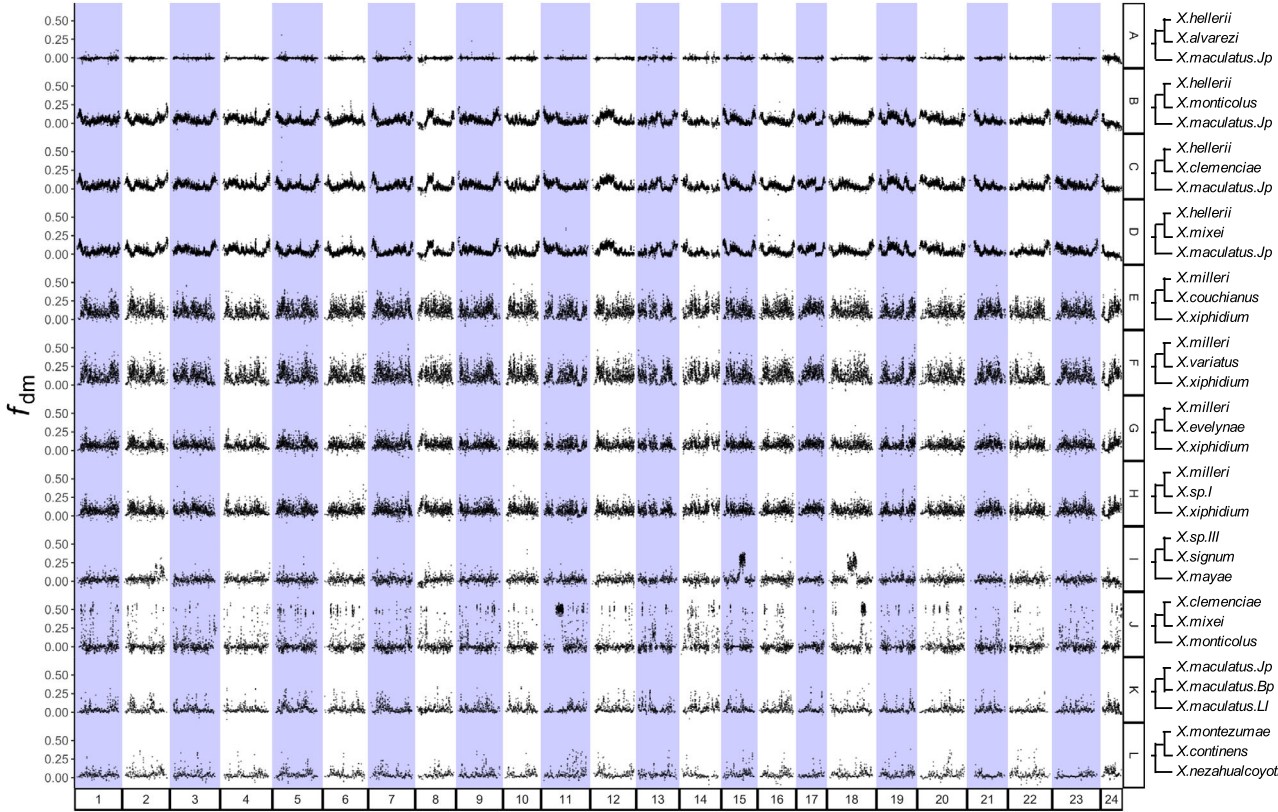

**Fig. 4 | Dot plots revealing genome regions derived from hybridization for species with or without hybrid history.** Trees on the right are in the format of ((P1, P2), P3), where the admixture signal between P2 and P3 was investigated. Values ($f_{dM}$) on the *y*-axis were calculated using Dsuite with modified $f_d$ statistics developed by Malinsky et al.[140], referring to the fraction of excess alleles in sliding-windows sharing between P2 and P3 than between P1 and P3, indicating hybridization origin of the genome region. Statistics were averaged over windows of 100 informative SNPs, moving forward by 50 SNPs. Row A refers to a genome without hybridization resources.

prior to the speciation of the swordless platyfish. Hence, the sword was lost in platyfish, but the females' preference for swords in platyfish remained. Such a phylogenetic pattern of gain and loss of the "sword" is not compatible with the pre-existing bias hypothesis which postulates that the sword evolved in ancestrally swordless species due to the preference of females for such a male ornament.

Mutant *mc4r* alleles are predicted to have a strong effect on the dynamics of growth and development. In *X. nigrensis* and *X. multilineatus*, *mc4r* is a major determinant of the age of puberty and the other characters linked to this trait (e.g., male size, ornamentation, courtship behavior, and territory establishment and defense)[34–36]. Breeding experiments have indicated that the timing of the onset of puberty is also heritable in *X. variatus*, *X. milleri*, *X. montezumae* and *X. cortezi*[31]. The absence of mutant *mc4r* copies in these genomes may be due to incomplete assemblies. However, a previous study showed that in *X. hellerii* the polymorphism in onset of maturity and the connected adult male size are not determined by the interaction of wildtype and mutant Mc4r isoforms[34]. Further, it was shown that large males have a much higher hypothalamic expression of *mc4r*, pointing to the same regulatory pathway but a different molecular mechanism underlying the same phenotypic polymorphism.

**Microevolution of *Xiphophorus* genomes at the genus level**

Genetic variation and genome evolution in vertebrates were investigated widely either within populations or across distant species, yet genome comparisons between species within a genus that diverged only a few million years ago are rare. According to earlier studies[8,47], *Xiphophorus* species radiated roughly around 5 MYA. Our study now

provides complete genomic resources for these closely related species to study genomic variation and evolutionary patterns over short evolutionary timescales.

We aligned all genomes to that of *X. maculatus*, which is assembled on chromosome-scale based on short and long-read data collection, including a meiotic map[49,75]. The alignments between chromosome-level assemblies revealed a high degree of conserved synteny. Moreover, in two species of *Xiphophorus* it was found that they have conserved recombination maps[76]. This may indicate that patterns of constraint and background selection may be generally shared across the genomes of these species. Indeed, we identify shared patterns of variation in diversity and divergence between species (Supplementary Fig. S1). Similar distributions of these highly diversified regions were also observed when the genomes were aligned to that of *X. hellerii* (Supplementary Fig. S2). When mapping homology across six species with chromosome-level genomes, including the outgroup *Priapella*, we found inversions and translocations of chromosome segments in platyfishes, despite conserved local synteny over the whole genome having been maintained for at least 25 million years (Supplementary Fig. S6). These bursts of rearrangements in the platyfish clade are striking and may be linked to a higher lineage-specific structural turnover of the karyotype. Future research on those rearrangements may aid in evaluating the impact of chromosome structure on speciation (see[77,78]) in the genus *Xiphophorus*.

It is worth noting that the phylogenomic results revealed larger genetic distances between strains of *X. maculatus* than among some species, particularly *X. meyeri*, *X. gordoni* and *X. couchianus*, or *X. multilineatus*, *X. nigrensis* and *X. pygmaeus* (Fig. 1). Species in

*Xiphophorus* have been usually identified and defined by differences in morphology, physiology, lifestyle and ecological adaptations. Intuitively, with more pronounced phenotypic differences, stronger interspecific genetic variation may be expected. However, our genomic data show a case where there is more genetic variation between populations of *X. maculatus* than between phenotypically divergent species. It will be interesting to investigate whether there is cryptic isolation between these *X. maculatus* populations using crossing experiments and tests for assortative mating. It will also be important to explore whether other factors, such as higher mutation rates or larger historical population sizes could have contributed to these patterns. Lineage-specific bursts of trait evolution vs. stasis linked to fast or slow diverging genomic features have been observed and well-studied between distant and long diverged lineages[79]. *Xiphophorus* offers the opportunity to investigate this now at a microevolutionary scale.

The size of certain gene families also varies significantly across the genus *Xiphophorus* (Supplementary Data 7). While these patterns have been previously reported for *paox* and *mc4r*[49], we identify exciting new patterns of gene family expansion in our genome-wide dataset. Perhaps most notable, given the importance of sexual selection in *Xiphophorus,* is the identification of diversification in gene families associated with functions in vision and olfaction. These gene families may allow the divergent adaptations among different species, for instance, in olfactory preferences[80], which are crucial premating barriers in *Xiphophorus* species[81,82]. The pronounced lineage-specific dynamics of gene family size provide new clues for trait diversification in the genus.

## Phylogenomics and hybridization

Our phylogenomic analyses divided *Xiphophorus* into three monophyletic groups: platyfishes, Northern swordtails and Southern swordtails, in agreement with previous studies on smaller subsets of species[13,15]. With the complete dataset, we were able to resolve the placement of several species that were previously disputed. For example, *X. xiphidium* was placed at the base of the platyfish clade in a phylogenetic tree constructed by using eleven nuclear loci[12]. Our results suggest that this is not correct and may be caused by restricted nuclear data available or previously undetected hybridization event(s) in the species history of *X. xiphidium* (Fig. 3). Similarly, gene flow between *X. signum* and *X. mayae* (Fig. 3) may have contributed *to X. signum* being identified as the sister lineage to *X. alvarezi* in previous work[15]. Together with another recent study[65], we consider mistyping of material for *X. continens* in previous studies as a cause for incorrect results[13,15].

Discordance between mitochondrial and nuclear phylogenies can indicate instances of past gene flow. In our phylogeny based on the complete mitochondrial genome, the platyfishes were grouped with Southern swordtails, but placed with Northern swordtails in the nuclear tree. A hybrid origin of platyfishes is a likely first proxy explanation. However, genomic segments shared between platyfishes and Southern swordtails (~6.5%) are not significantly more than those shared between Northern swordtails and Southern swordtails (~4.7%), hence ILS cannot be excluded to account for this mito-nuclear discordance.

The mitochondrial genomes of two Southern swordtail species, *X. clemenciae* and *X. monticolus*, are perfectly nested within the platyfish clade, prompting us to revisit our earlier hypothesis that they may have arisen through hybridization[11–14]. Past research proposed that this mito-nuclear discordance could be generated by ILS since whole genome sequences did not support admixture in the nuclear genome[15,83]. Here, using *P. lacandonae* as the outgroup, we demonstrate that the amount of potential hybridization-derived loci of *X. clemenciae* is significantly higher in the nuclear genome than expected for ILS alone (Supplementary Fig. S15B, D), hence, indeed arguing for a hybrid origin. A potential disagreement with past research may be explained by results from our study that this hybridization event appears to be older than the divergence of platyfishes and Northern swordtails - more ancient than previously thought (Fig. 3).

Our result also suggests that the hybridization resulting in the origins of *X. clemenciae* and *X. monticolus* is the same event that occurred before the split of the *X. clemenciae*, *X. monticolus* and *X. mixei* clade. Different from the other two species, *X. mixei* has its mitochondrial haplotypes assigned to the Southern swordtails. Multiple independent fish sampling in the field from previous studies have implied that *X. monticolus* and *X. clemenciae* harbor mitochondria of only the "minority" parental species[11–13]. Thus, the presence of Southern swordtail mitochondria in *X. mixei* may be explained by backcrossing of hybrid fish with a female Southern swordtail (Fig. 3), in agreement with the retained female preference for swords in platyfish/swordtail hybrids[11].

More generally, the now complete genomic resources of all *Xiphophorus* species provide us with the opportunity to establish a broader view of the hybridization history among all current and ancestral lineages of this genus. Our pan-genus genomic information uncovered the common ancestral hybridization event in the lineage of *X. clemenciae*/*X. mixei*/*X. monticolus* was contributed by an ancestor of platyfishes and Northern swordtails (Fig. 3), suggesting that the event that led to mitonuclear discordance was much more ancient than previously hypothesized (Supplementary Fig. S16). Our phylogenomic analysis clearly reveals that the hybridization event preceded the formation of the three Southern swordtail species. In addition, we also saw hybridization between *X. nezahualcoyotl* and *X. continens* (Fig. 3)[65], which turns out to be more prominent than the previously reported hybridization between *X. nezahualcoyotl* and *X. cortezi*[15,16], making this lineage of particular interest for unraveling a complex history of hybridization in that group. Finally, our study discovered ancient hybridizations between *X. xiphidium* and the ancestor of *X. gordoni, X. meyeri, X. couchianus, X. variatus, X. evelynae,* and *X. sp* I (Fig. 3), unveiling a profound influence of *X. xiphidium* on the evolution of platyfishes through hybridization.

## Genome stabilization after hybridization

Homozygous regions in hybrid genomes evolved through recombination events in successive generations and were finally fixed to one or the other part of the parental ancestry, a process referred to as "genome stabilization"[84]. We refer to the ancestry from the "minor parent" as hybridization ancestry or foreign ancestry.

We found hybridization ancestry dispersed in different genomic areas following recent (Fig. 4I–L) and ancient (Fig. 4B–H) hybridization events. We failed to detect common enriched functions for foreign PCGs that were fixed in different hybridization events. This may reflect different selective forces and adaptations during the stabilization of different hybrid genomes or distinct outcomes of genetic drift in these independent hybridization events in their respective environments that are likely to differ in a number of biotic and abiotic parameters.

Different from the other species with hybridization history, *X. signum* exhibits a divergent pattern: while prevalent across chromosomes in other hybrid genomes, admixed signals are only steeply clustered, especially on chromosomes 15 and 18 in *X. signum* (Fig. 4I). The retention of only a small fraction of hybridization ancestry may be due to a limited adaptive value of the minority parent genome or structural features, like inversions. For zoogeographical reasons, a recent hybridization of *X. signum* with *X. mayae* as the causation of this pattern can be excluded since their distributions do not overlap. Small clusters of hybridization ancestries were also observed on chromosome 11 and 18 of *X. mixei* (Fig. 4J). These clusters may contain functionally linked loci whose separation will be purged by selection against incompatibility. Alternatively, recombination events merely failed to break up these regions.

Based on the whole-genome analysis, we detect hybridization in the lineage leading to *X. clemenciae*, *X. monticolus* and *X. mixei* (Fig. 4B–D). As expected from a shared ancestral hybridization event, we find that patterns of local ancestry are highly concordant across the genomes of these three species. This pattern is also observed in hybridization events involving *X. xiphidium* (Fig. 4E–H). Hybridization of different ages have been noted also for hyperdiverse clades such as cichlid fish, however the derived species possess different combinations of biparental ancestry. This can be explained as being at least partly due to the rapid radiation of cichlid fish by sorting various combinations of biparental ancestry into different species[85–87]. Less rapid radiation in Southern swordtails may explain the stabilization before speciation in our case.

The molecular evolution of hybridization-derived regions has been studied by comparing the orthologous regions between two parental species, hence reflecting their evolution before the hybridization event[16]. As a result of hybridizations in ancestral lineages, common hybridization-derived regions shared between species provide us with an opportunity to investigate their molecular evolution after hybridization. We found a significantly lower divergence and slower nonsynonymous substitution rate in hybridization-derived regions (Supplementary Fig. S15C, E). This is congruent with the hypothesis that genomic regions with lower substitution rates, particularly at nonsynonymous sites, might be less likely to be associated with negative epistasis and incompatibilities between species[88–90]. It is worth noting that an opposite association was found in cases of recent hybridizations that occurred less than a few hundred generations ago[76]. We also detected a slightly higher value of dN/dS for coding genes with hybrid ancestry (Supplementary Fig. S1F). This could be explained by less constraint on such regions.

Our whole genome analysis uncovered that hybridization had occurred frequently during the evolution of *Xiphophorus* and clearly played an important role in speciation, phenotypic evolution, and local adaptation. Genome stabilization is a hallmark of the molecular evolution of such hybrid-derived genomes. Whether the conservation of ancestral regions is a cause or consequence of divergence and speciation requires more detailed studies on lineage-specific adaptations in the 26 currently known species.

## Methods
### Ethical statement
All individuals used for whole genome sequencing were raised at the fish facilities of the Biocenter of the University of Würzburg and the *Xiphophorus* Genetic Stock Center at Texas State University following approved experimental protocols through an authorization (568/300-1870/13) of the Veterinary Office of the District Government of Lower Franconia, Germany, in accordance with the German Animal Protection Law (TierSchG) and with an approved Institutional Animal Care and Use Committee protocol (IACUC 7381). Texas State University has an Animal Welfare Assurance on file with the Office of Laboratory Animal Welfare, National Institute of Health (A4147).

### Sampling and genome sequencing
High molecular weight DNA was prepared from pooled soft organs of single individuals by a phenol/chloroform extraction procedure[91]. For *P. lacandonae* and *X. maculatus* Bp, muscle samples were used to obtain high molecular weight gDNA using the QIAGEN MagAttract HMW DNA kit (QIAGEN, Germantown, MD, USA) according to the manufacturer's protocol. The quality and quantity of the gDNA were analyzed using a 5400 Fragment analyzer (Agilent Technologies, CA, USA) and Qubit 2.0 Fluorometer (Invitrogen, Life Technologies, CA, USA). DNA libraries were generated using the 10x Genomics Chromium technology according to the manufacturer's instructions. Gel Bead-In-Emulsions (GEMs) were created from a library of Genome Gel Beads combined with 1.5 ng of gDNA in a Master Mix and partitioning

oil, using the 10x Genomics Chromium Controller instrument with a micro-fluidic Genome chip (PN-120257). The GEMs were then subjected to an isothermal incubation step. Bar-coded DNA fragments were extracted and underwent Illumina library construction, as detailed in the Chromium Genome Reagent Kits Version 2 User Guide (PN-120258). Library yield was measured through the Qubit dsDNA HS assay kit (Thermo Fisher Scientific, Waltham, MA, USA). Library fragment size and distribution were measured using an Agilent 2100 Bioanalyzer High Sensitivity DNA chip (Santa Clara, CA, USA). The DNA was sequenced on a NovaSeq with a $2 \times 150$ bp read metric.

For *P. lacandonae*, muscle tissue from the same individual was used to construct a Hi-C chromatin contact map to enable chromosome-level assembly. Tissue fixation, chromatin isolation, and library construction for Hi-C analysis were performed according to the manufacturer's instructions (Dovetail Genomics, Chicago, USA)[92]. After checking the insert size, concentration, and effective concentration of the constructed libraries, the final libraries were sequenced using the Illumina Novaseq platform (San Diego, CA, USA) with a 150-bp paired-end strategy.

### Genome assembly
For genomes sequenced as Illumina short reads, we assembled each genome in two parallels and chose the result with higher completeness or continuity. First, contigs were assembled using SOAPdenovo (Version 2.04)[93], scaffolded using SSPACE (version 3.0)[94], and arranged into chromosomes using cross_genome[95]. In parallel, contigs were assembled and scaffolded using Platanus (version: 1.2.4)[96], and then arranged into chromosomes using cross_genome.

SOAPdenovo assembles scaffolds by running de Brujin graph to merge all read-subsequences of length "k-mer". We tried different k-mers (from 75-mer to 105-mer), chose the result with the longest N50 scaffold length, and then filled the gaps using GapCloser (version 1.12)[96]. The gap-filled scaffolds were then broken into contigs again at the N-linked positions. We removed those contigs that aligned to the mitochondrial genome and then rescaffoled them using SSPACE with parameters: -x 0 -z 200 -g 2 -k 2 -n 10.

Given the highly conserved synteny among genomes of Cyprinodontiformes[54], we decided to assemble the scaffolds into chromosomes based on cross-species synteny using cross_genome. Highly qualified genomes of *X. couchianus*, *X. maculatus*, *X. hellerii,* and *X. malinche,* respectively, were used as the reference for Northern Platies, Southern Platies, Southern Swordtails and Northern Swordtails.

Platanus works well for assembling divergent heterozygous regions. Contigs were first assembled using the command "Platanus assemble" with parameter "-t 21 -m 80". Contigs that aligned to the mitochondrial genome were removed and the rest were scaffolded using the command "platanus scaffold". We then filled the gaps in scaffolds using "Platanus gap_close" and GapCloser and arranged the scaffolds into chromosomes using cross_genome.

Genomes sequenced as 10X linked-reads were assembled using Supernova (version 2.1.1)[97] with parameters: –maxreads 298666666 –local cores = 8. The assembly was output in style "pseudohap2" where two 'parallel' pseudohaplotypes were created and placed in separate FASTA files. From the two consensus assembly results, we chose the one with higher completeness or continuity.

For *P. lacandonae*, Hi-C reads were mapped to the draft 10X genome assembly using HiC-Pro (v. 2.8.0) with default parameters[98].

The completeness of each assembly was estimated using BUSCO (version 2.0.1)[99] by the completed partition of the actinopterygii_odb9 database, which contains the conserved gene set across Actinopterygii ($n = 4584$). The completeness could also be evaluated by comparing the estimated genome size with the actual size of the sequences. The continuity of the assembly was evaluated by N50 scaffold length, which was calculated by the Perl script assemblathon_stats.pl[100].

## Genome annotation

Genomes were annotated using a further developed and improved pipeline from our previous studies[101]. In brief, the assembly was masked at repeat regions and then mapped with protein sequences and RNA reads to collect homology and transcriptome gene evidence. Consensus gene models, given their high quality, were used to train the de novo gene predictor: AGUSTUS (version 3.2.3)[102], which was then run with all collected gene evidence as hints to collect ab initio gene evidence. The final set of gene annotations was generated by synthesizing all three types of gene evidence.

In detail, to identify and mask repeats from the genome, we first screened the assembly for automated discovery of transposable elements using RepeatModeler[51]. The consensus sequences of repeats were used as a de novo repeat library, together with Repbase and FishTEDB[103]. They were transferred to RepeatMasker for repeat identification and masking.

To collect homology gene evidence, we downloaded 455,817 protein sequences from the vertebrate database of Swiss-Prot (https://www.uniprot.org/statistics/Swiss-Prot), RefSeq database (proteins with ID starting with "NP" from "vertebrate_other") and the NCBI genome annotation of human (GCF_000001405.39_GRCh38), zebrafish (GCF_000002035.6), platyfish (GCF_002775205.1), medaka (GCF_002234675.1), mummichog (GCF_011125445.2), turquoise killifish (GCF_001465895.1) and guppy (GCF_000633615.1). These protein sequences were aligned to the assembly using Exonerate (https://www.ebi.ac.uk/about/vertebrate-genomics/software/exonerate) and Genewise[104], respectively, to predict gene location and intron/exon structures. To speed up Genewise, GenblastA was used a priori to find the rough location of the alignment on the assembly[105].

To collect transcriptome gene evidence, we downloaded RNA data from SRA NCBI for *X. andersi, X. clemenciae, X. evelynae, X. gordoni, X. monticolus, X. multilineatus, X. nezahualcoyotl, X. nigrensis, X. pygmaeus, X. signum, X. variatus, X. xiphidium, X. mayae, X. meyeri,* and *X. milleri*. RNA of *P. lacandonae* and *X. maculatus* was isolated from separate or mixed tissues using TRIzol Reagent (Thermo Fisher Scientific) according to the supplier's recommendation and sequenced in 150-bp paired-end reads using the BGISEQ platform. RNA data from a previous study was used for *X. birchmanni* and *X. malinche*[42]. For species without RNA data, the resource from closely related *Xiphophorus* species was used for the annotation. All RNA-seq reads were cleaned using fastp[106], and were mapped onto the assembly using HISAT[107]. The resulting bam file was then interpreted by StringTie for gene locations and structures[108]. In another parallel, transcript sequences were assembled based on the bam file using Trinity[109]. We then aligned these transcript sequences to the assembly for gene prediction using splign[110].

AUGUSTUS was used for collecting de novo gene evidence. We first trained AUGUSTUS using BUSCO with the parameter "-long"[99]. In addition, those genes that were predicted repeatedly by Exonerate, Genewise, StringTie, and Splign were considered to be of high quality and were used to train AUGUSTUS for the second round. The trained AUGUSTUS was then run on the assembly with all homologous and transcriptome gene evidence as hints for an ab initio gene prediction.

## Identification of orthologous genes

Orthologous genes were identified by sequence-similarity clustering followed by gene tree reconstruction within each cluster. Protein sequences of the longest transcript of each gene were pooled together and passed through an all-versus-all BLAST. Then, for each two genes, an H-score was calculated from the BLAST score to index the sequence similarity[111,112], so the genes could be clustered using Hcluster_sg with *P. lacandonae* as the outgroup[113]. Within each cluster, a gene tree was reconstructed using TreeBeST v.0.5 https://github.com/lh3/treebest. Based on this, the orthologous genes between species were typed as "*n* to *m*" orthology (*n* and *m* are positive integers; there are cases where *n* = *m*).

## Size dynamic of gene families

Gene families were identified by clustering genes with similar protein sequences (see "Method" section: identification of orthologous genes). For each gene family, we counted its member numbers in each species. Then CAFE5 was used to retrieve gene families that had been significantly expanded or contracted during the lineage evolution[114].

## Mitochondrial genome tree

The mitochondrial genomes were assembled from the primary reads using MITObim and norgal[115,116]. MITObim assembles the genomes by baiting the reads and iteratively mapping them on a reference mitochondrial genome. The mitochondrial genomes of *X. couchianus, X. maculatus, X. hellerii,* and *X. malinche* were downloaded from NCBI and used as the reference for Northern Platies, Southern Platies, Southern Swordtails and Northern Swordtails. For comparison and complementation, we also used norgal for de novo assemblies. As input for both assemblers, 10% of clean reads were sampled from the whole-genome sequencing (WGS) using BBmap[117].

Mitochondrial genomes were annotated using the MitoAnnotator pipeline of the Mitofish database[118]. DNA sequence alignment of each protein-coding gene, rRNA, and tRNA was made using MACSE for coding sequences and MUSCLE for non-coding sequences[119,120]. We manually screened the alignments to check for potential assembly or annotation errors, which were then curated by selecting a better assembly from MITObim and Norgal results. With respect to annotation errors in protein-coding genes, we selected the better assembly, and retrieved the coding sequence by aligning homologous protein sequence on the DNA sequence using Genewise. After curation, alignment gaps were removed using Gblocks[121].

The phylogenetic tree based on mitochondrial genomes was built using RAxML and MrBayes, respectively[122,123]. RAxML infers phylogeny relationships under maximum likelihood. As input, we concatenated the DNA alignments of the 13 protein-coding genes, 2 rRNAs, and 22 tRNAs into a single giant alignment. In addition, the region information of each gene was passed to RAxML in a partition file for the partitioned analysis. This resulted in an 15,515 bp long alignment with partitions as follow: ATPase-6 (1-681), ATPase-8 (682-846), COIII (847-1629), COII (1630-2319), COI (2320-3876), Cyt-b (3877-5016), ND1 (5017-5976), ND2 (5977-7020), ND3 (7021-7365), ND4L (7366-7662), ND4 (7663-9042), ND5 (9043-10869), ND6 (10870-11382), 12S-rRNA (11383-12319), 16S-rRNA (12320-13969), tRNA-Ala (13970-14038), tRNA-Arg (14039-14107), tRNA-Asn (14108-14180), tRNA-Asp (14181-14251), tRNA-Cys (14252-14315), tRNA-Gln (14316-14386), tRNA-Glu (14387-14455), tRNA-Gly (14456-14526), tRNA-His (14527-14596), tRNA-Ile (14597-14663), tRNA-Leu2 (14664-14735), tRNA-Leu (14736-14809), tRNA-Lys (14810-14882), tRNA-Met (14883-14950), tRNA-Phe (14951-15018), tRNA-Pro (15019-15087), tRNA-Ser2 (15088-15156), tRNA-Ser (15157-15227), tRNA-Thr (15228-15300), tRNA-Trp (15301-15373), tRNA-Tyr (15374-15443), tRNA-Val (15444-15515). The analysis was performed under the General Time Reversible (GTR) + Gamma phylogenetic model with 1000 rapid bootstrap tests.

MrBayes infers phylogeny using Bayesian methods. The concatenated alignment was also used as the input data. We ran the inference under a General Time Reversible model with a proportion of invariable sites and a gamma-shaped distribution of rates across sites by setting the parameter "lset nst = 6 rates = invgamma". Three runs starting from a different tree were launched with six chains for each running 50 million generations. The results were sampled each 1000 generations and the first 25% samples were discarded as burn-in.

## Phylogenomic tree based on whole genome alignment (WGA)

The multiple WGA across all species was built by merging pairwise genome alignments of each species to *X. maculatus*. We used minimap2 for the pairwise genome alignments with parameter "-cx asm20 −cs = long"[124]. The alignments were then refined using Genome

Alignment Tools from Hiller lab as follows[125]: alignments were first chained up using axtChain, and the unaligned loci flanked by aligning blocks were then re-aligned using patchChain.Perl, newly-detected repeat-overlapping alignments were incorporated into the alignment chains using RepeatFiller. To improve the specificity of alignments, the obscure local alignments were detected and removed using chain-Cleaner. We then used chainNet to collect alignment chains hierarchically to capture only orthologous alignments. Finally, pairwise genome alignments were merged into the multiple WGA using MULTIZ[126]. The final WGA contains 340,357 alignment blocks with a mean length of 1214 bp, the longest of ~34 kb, and the sum of ~413 Mb, spanning ~58% of the genome.

To reconstruct a phylogenomic tree based on the WGA, we removed alignment blocks shorter than one kb to avoid potential false-positive-alignments. The remaining alignment blocks were then trimmed using trimAl and then concatenated into a ~342 Mb long alignment[127]. The alignment was passed to RAxML for a maximum likelihood inference of the phylogenomic tree under the GTR + Gamma model with 1000 rapid bootstraps. The bootstrap values of nodes were all at 100%.

To construct a coalescent tree, we sampled 19,111 alignment blocks that are at least one kb long and 20 kb away from each other. From each of the blocks, we randomly cut out a one kb window to infer the maximum likelihood tree using RAxML under the GTR + Gamma model with 100 rapid bootstraps. To reveal the discordance across the resulting locus trees, we selected the top 10,000 trees with the highest total bootstrap value and aligned them in DensiTree[128]. Before constructing the coalescent tree, we filtered these locus phylogenies by removing trees with none of the nodes supported over 75% and collapsing nodes with bootstrap value <75%. The final coalescent tree was built based on the filter trees using weightedASTRAL[63,64]. In an additional attempt, we used a supporting value of 10% as the cutoff for tree filtering before constructing the coalescent tree. This resulted in the exact same topology. We also used the CASTER site for the coalescent tree reconstruction. CASTER-site uses whole genome alignment directly to identify the site patterns without demarcating loci. This spared the conflicting requirements to use longer loci and the need to avoid recombination within loci but maintained abundant recombination across loci, and prevented the pitfalls of gene tree estimation errors[64].

We also calculated site concordant factors (sCF) for both concatenation- and coalescence- based trees to reveal the disagreement among sites using IQ-TREE 2 with the parameter "--scf 100 -seed 438607"[129].

## Phylogenomic tree based on protein coding genes
3259 one-to-one orthologous genes were identified across all species based on sequence similarity followed by synteny confirmation. With these orthologies, we reconstructed the phylogenomic tree using RAxML with the concatenated protein sequences, coding sequences, and 4DTV (fourfold degenerate sites) sequences, respectively. Evolutionary model was automatically searched for protein sequences with parameter "-m PROTGAMMAAUTO", while set to GTR+Gamma for coding and 4DTV sequences. With coding sequences, analyses were partitioned by codon position. Node confidence of each tree was assessed by 200 bootstrap replicates.

## Phylogenomic tree construction based on conserved non-coding elements (CNEs)
CNEs were retrieved from whole genome alignments (WGA) by first identifying conserved regions using phastCons and then removing coding regions[130]. Given the close phylogenetic relationships among *Xiphophorus* species, we let phastCons itself estimate the phylogenetic models directly from the data, using an "unsupervised" learning algorithm. The initial non-conserved model was calculated from fourfold degenerate sites using phyloFit[130].

For phylogeny reconstruction, similar concatenation and coalescence methods were used for CNEs as to WGA (see Method section Phylogenomic tree based on whole genome alignment). Except when building loci trees for coalescence analysis, instead of cutting out one kb-long alignment windows, we binned CNEs into 100 kb slide windows, then concatenated the alignments and built the locus tree for each window independently. This change is due to the short length of and the lack of sufficient signals in each independent CNE.

## Divergence time estimation
MCMCTree was used to infer divergence time under a relaxed-clock model (correlated molecular clock) with approximate likelihood calculation and maximum likelihood estimation of branch lengths performed[131]. First, baseml roughly estimated the substitution rate based on the coding sequence alignment and phylogenetic tree[132]. The substitution model was determined using modelgenerator.jar[133]. Second, mcmctree ran for the first time to estimate the Gradient and Hessian. The result was output into a file out.BV and then used for the final run of MCMCTree to perform approximate likelihood calculations. Third, the final Markov chain Monte Carlo process was run for 550,000 steps. The first 50,000 steps were discarded as burn-in. Afterward, 10,000 samples were collected by sampling every 50 steps. As time calibrations, we set the root in between 25 and 26 MYA and diving of Xiphophorus in between 4 and 6 MYA.

## Inference of hybridization history
The whole genome alignments of four target species were constructed using minimap2 and MULTIZ, then a one kb-long window was extracted from each of the alignment blocks that are at least 1 kb long and 20 kb away from each other. For each window, we applied the AU-test to select the optimal tree topology from the three candidates. The site-wise likelihoods were first calculated using RAxML, and then the AU-tests implemented in Consel 0.20[134]. At last, trees with an AU P-value higher than 0.95 were included for the count and comparison.

We also applied Patterson's D-statistic on these alignment windows to detect the gene flow between two species. Patterson's D-statistic is also known as ABBA-BABA statistic, where alignment sites ABBA refers to those at which species 2 and 3 share a derived allele "B", while species 1 has the ancestral state "A", as defined by the outgroup: species 4. Likewise, BABA refers to those sites at which species 1 and 3 share the derived state. The counts of the two type sites would equal each other when there is no gene flow. This null hypothesis was tested using a two-sample z-test. The standard error (SE) of value D was determined using the jack-knife resampling method[135].

AU-test and the *D*-statistic can infer the hybridization history when only three species were targeted. To calculate the *D* and *f4*-ratio statistics across all combinations of the studied species, we also implemented Dsuite[66]. At last, specific introgressed loci were retrieved with the parameter "Dinvestigate -w 50,5" using the *f*dM index, which may flaw in simulations, but it is the best option still for our analyses.

We also used SNaQ to detect hybridizations[136]. Given the intensive computation to include all taxa, we selected fewer taxa to confirm the hybridization events that happened in the ancestor lineage of the *X. monticolus*, *X. clemenciase* and *X. mixei* clade; and in the *X. xiphidium* lineage. For both calculations, we used the 19,111 locus trees as input to count the quartet CFs. Then SNaQ was implemented with hm = 0 to 3 and 10 runs each. The resulting trees were shown by Dendroscope3[137].

## Retrieval of sequences for *xmrk* and *egfrb*
First, we retrieved the sequences of *xmrk* and *egfrb* for each species from the genome assembly through the result of genome annotation and *ab inito* homology search. Then for those species with no *xmrk* found in the assembly, we made an effort to recover it directly from the reads. Protein sequences of *xmrk* from *X. maculatus* (XP_023181635.1 and XP_023185812.1) were used as the reference. Reads were aligned to

the reference using diamond[138]. We then assembled the retrieved reads using cap3 and translated the resulting contigs into protein sequences using GeneWise[104,139]. At last, for each exon of a reference, the contig matches the best was kept and assembled into the final sequence.

## PCR and discriminatory digestion of *xmrk* and *egfrb*

Due to the high degree of conservation of *xmrk* and *egfrb* coding sequences it was impossible to design primers from coding exons that would discriminate between both genes and produce gene-specific amplicons. On the other hand, primers designed in non-coding regions are not versatile across multiple species. To solve the problem, we used the PCR process with common primers followed by a discriminatory restriction enzyme digestion of *xmrk* and *egfrb*.

We designed primers that target an exon of *xmrk* but are also able to amplify the homologous region from *egfrb*. Then with a restriction enzyme that cuts the PCR products of *egfrb* but not *xmrk*, we confirmed the presence of *xmrk* by observing PCR products resistant to the enzyme. On the other hand, the absence of *xmrk* was confirmed by observing all PCR products broken down into two fragments.

Total DNA was extracted from a fin clip obtained from adult animals using the DNeasy Blood & Tissue Kit from Qiagen, following the manufacturer's instructions. PCR amplification was conducted employing GoTaq® Green Master Mix (Promega). The reaction mixture, with a total volume of 25 µL, comprised 0.5 µL of 10 µM forward primer (5′-CAGGTGGATGGCAGGTGTG-3′) and 0.5 µL of 10 µM reverse primer (5′-GAGCAGCGCCACAATTACAG-3′), 12.5 µL of GoTaq® Green Master Mix, 9.5 µL of nuclease-free water, and 2 µL of DNA template. The PCR conditions included an initial denaturation step at 95 °C for 2 min, followed by 35 cycles of denaturation at 95 °C for 30 s, annealing at 61.5 °C for 1.5 min, and extension at 72 °C for 2 min, with a final extension step at 72 °C for 10 min. Subsequently, the PCR product (1 µg) was digested using SphI-HF (New England Biolabs) for 1 h at 37 °C. Finally, the PCR and digested products were subjected to analysis via electrophoresis on a 0.9% agarose gel.

## Reporting summary

Further information on research design is available in the Nature Portfolio Reporting Summary linked to this article.

## Data availability

Raw reads of the whole genome sequencing generated in the study have been deposited in SRA under accession number PRJNA972672. Assemblies and annotations are available in figshare under https://doi.org/10.6084/m9.figshare.23596515.v1. Assemblies of *X. maculatus, X. couchianus, X. helleri, X. birchmanni,* and *X. malinche* are from previously published sources. They are available in GeneBank under accession number GCA_002775205.2, GCA_001444195.3, and GCA_001443345.1 or figshare under https://doi.org/10.6084/m9.figshare.23596515.v1.

## Code availability

The source code used in the manuscript can be obtained from https://github.com/dukecomeback/XiphoMicroEvo.

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

## Acknowledgements

We thank the *Xiphophorus* Genetic Stock Center for providing several fish samples used in this study. This work was supported by a grant from Texas State University to MS and an ERC advanced grant (293700) to A.M. The work was further supported by NIH R24OD031467, NIH R15CA223964, NIH 1R35GM133774 to Y.L. and M.S. and CPRIT RP200657 to M.S. Computational work associated with the study was performed on the Learning, Exploration, Analysis, and Processing (LEAP) next-generation High-Performance computing cluster at the Texas State University, San Marcos, TX.

## Author contributions

Ma.S. and A.M. designed the study and wrote the paper. K.D. performed the assemblies, annotations, all bioinformatic analyses and drafted the manuscript. H.P. sequenced and assembled the genomes of *P. lacandonae* and *X. maculatus* Bp. R.B.W., W.C.W., Mo.S., Y.L., T.O.D., and A.M. provided raw data and assembled genomes. J.M.B.R. performed the PCR experiments. M.G.O. compared phylogenomics results and revised the manuscript. All authors interpreted the results and participated in manuscript writing.

## Funding

## Competing interests

The authors declare no competing interests.
