## [Peer Review File · Nature Communications]

Phylogenomics analyses of all species of swordtail fishes (Genus *Xiphophorus*) show that hybridization preceded speciation

Corresponding Author: Professor Manfred Schartl

Version 0:

Reviewer comments:

Reviewer #1

(Remarks to the Author)

This study is a superb piece of work that leaves few stones unturned.

All elements of a good study are present:

The study system is interesting, complex, well-known yet still much to be discovered, with many important outstanding questions.

The sampling is excellent, all species are included

The approach is outstanding, full genomes are used

Thus, the setup is impeccable.

What I most appreciate is that studies that include a lot of genomes usually restrict their analyses to one topic: phylogeny, or one specific gene.

This study, instead, answers a lot of questions. The title, and the main goal, is to answer a fundamental evolutionary question, hybridization. This is a choice, and in my opinion a good one.

Meanwhile, a number of other very essential issues are dealt with as by catch from this study: melanoma (a topic that has been core to *Xiphophorus* research for decades) etc...

Overall, a huge pleasure to read this great paper.

Well written, nothing to change in my mind.

(and I don't think I have said that about reviewed papers in over a decade)

Reviewer #2

(Remarks to the Author)

-For a paper focused on phylogenomics, I find the phylogenomic analyses to be rather superficial. When examining the nDNA genomic dataset, differences arise between the concatenation ML and the Astral trees. However, it remains unclear what the primary driver of discordance is. Is it ILS? ("Discordance between gene trees and species trees can be generated by the coalescent process or by historical gene flow. We investigated this further using coalescent methods to build a nuclear phylogeny.") Authors should do more in depth analyses, looking at the sensitivity of the reconstructions. Additionally, since genomes are annotated, I anticipated seeing not only a protein-based tree but also trees based on CNNEs and UCEs (both concatenation ML and coalescent). Without more extensive analyses, it is challenging to identify the source of the incongruence.

Most importantly, given the pervasive introgression in the group (Fig. 2), why not run a set of analyses that exclude all introgressed loci? Such analyses might shed light on the sources of discordance regarding the unstable placement of several species, as well as the mitonuclear discordance reported (see below).

The coalescent nDNA trees (and other trees that should be estimated as suggested above) should be included in Fig. 1 as well. A brief discussion about the incongruent results obtained with Astral concerning *X. clemenciae*, *X. monticolus*, and *X. mixei* should be added, as it renders the southern swordtail clade non-monophyletic. Is it an artifact arising from gene tree error, ILS, or is the introgression pulling clemen + mont closer to the platyfish? Additionally, the Astral tree is slightly less discordant with the mtDNA tree than the concatenation tree (perhaps again due to the introgression?).

-I also wonder why no phylogenetic network analyses are presented, considering in particular that *Xiphophorus* is a model

clade (if not the poster child) for reticulation. See <https://journals.plos.org/plosgenetics/article?id=10.1371/journal.pgen.1005896>. While I understand that phylogenetic networks can be computationally intensive to run, an alternative could be to use multiple independent subsets with fewer loci and fewer taxa.

-Did the authors include partitions in their phylogenetic analyses? It is less straightforward to do so with alignment windows/blocks that can include a plethora of locus type (exons, introns, flanking regions, UCEs, etc). But for protein coding genes, analyses should be partitioned minimally by codon position.

-“Two studies found a close association of *X. continens* with *X. pygmaeus* 13,14 while all our trees place *X. continens* as sister taxon to *X. montezumae*. Our result is consistent with another recent study on Northern swordtails 55” Please comment on the source of discordance. Is it a miss ID in previous studies? (see also below).

-Genome annotation: you have a Table S6 reporting stats after annotations for all genomes, but it isn’t intuitive for the reader to go there. Summarize the results from all those supplementary tables somewhere. I don’t understand why simple things like building a mtDNA tree has so much detail, but other important approaches are barely explained in the main text (e.g., RNAseq).

-“The resulting phylogenetic estimates were almost identical to the WGA tree mtDNA tree nuDNA tree (Supplementary Fig. S8), except *X. xiphidium* and *X. alvarezii* 219 were placed in slightly different positions within their respective clade.” Mayae and kallmani also have unstable placements. Same goes for populations of *maculatus*. Please comment!

-“Our whole genome resequencing of the material used in the conflicting studies revealed mistyping of the material (data not shown). Our analysis also strengthens the position of the Northern platyfishes species as being a crown group within the Southern platyfishes. Consequently, a distinction between both groups is no longer justified and we refer to the whole clade as “platyfishes.” This statement has major taxonomic implications; the comparison with previous studies needs to be shown. Also, why choose “platyfishes” over “swordtails”? I think many more people are familiar with the latter common name (Google hits 725k for “platyfish” and 3.5M for “swordtail”).

-“The bootstrap values of nodes were all at 100%.” It’s quite obvious that bootstrap support in WG phylogenomic analyses are inflated. Authors should use a more reliable metric of nodal support, like gene concordant factors and/or site concordant factors.

-“Trees with none of the nodes supported over 75% were removed, nodes with bootstrap value <75% were collapsed. The final coalescent tree was built based on the filter trees using ASTRAL-II.” 75% seems like an overly stringent cutoff. The Astral tutorial recommends collapsing branches with less than 10% support: <https://github.com/smirarab/ASTRAL/blob/master/astral-tutorial.md>. Try another cutoffs to see if the topology changes.

-“*xmrk* melanoma oncogene. The *xmrk* gene was originally found in the Southern platyfish, *X. maculatus*. From the whole genome information of all species, we were able to retrieve *xmrk* sequences in a total of nine platyfish and Northern swordtail species, but not in any of the Southern swordtails. Due to the high degree of conservation of *xmrk* and *egfrb* (its proto-oncogenic precursor) coding sequences, several nodes of the gene tree are only poorly supported (Supplementary Fig. S6). Despite this, a monophyletic origin of *xmrk* is visible.” Given that most genomes were assembled using short reads, confirming absence would require PCR. Minimally, refrain from making a strong case about the gene absence in some species.

-“From the genome assemblies we were able to identify multiple mutant copies, which are predicted to be defective, also in *X. maculatus*, *X. xiphidium*, *X. birchmanni*, *X. malinche*, *X. evelynae* and *X. sp III*, but not in any of the Southern swordtails.” Genome assemblies can be compromised by sequencing error (in particular short-read genomes). Again, ruling out that option requires PCR.

-“To collect transcriptome gene evidence, RNA-seq reads from multiple tissues were cleaned using fastp96, and were mapped onto the assembly using HISAT” -> which species?

-“The size of certain gene families also varies significantly across the genus 403 *Xiphophorus* (Supplementary Table S7). While these patterns have been previously 404 reported for *paox* and *mc4r46*, we identify exciting new patterns of gene family expansion 405 in our genome-wide dataset. Perhaps most notable given the importance of sexual selection 406 in *Xiphophorus* is the identification of diversification in gene families associated with vision and olfactory functions.” There’s no way for the reader to see this. Table S7 contains all the raw data output from the Cafe analysis. This needs to be summarized somehow for ease of interpretation. Also, olfactory gene families are always expanded – that’s not something unique to swordfishes.

-“Here, we add twenty-nine additional genomes to these existing genomic resources, which were sequenced using Illumina, 10X, PacBio and/or Hi-C techniques in this study (Supplementary Table S1). This now provides a complete genome resource of all *Xiphophorus* species, including all 26 previously identified species (<https://www.ncbi.nlm.nih.gov/Taxonomy/Browser/wwwtax.cgi?id=8082>), three undescribed taxa, *X. sp I*, *X. sp II*, *X. sp III*, two new strains of *X. maculatus*, and a reference genome for the species *Priapella lacandonae* as outgroup.”

This is misleading. Most genomes were sequenced using Illumina short reads (shotgun genomes). As written, the ms. gives

the impression that most new genomes are chromosome level. Clarify! State for example, "we add twenty-nine additional genomes to these existing genomic resources, including XX chromosome level genomes (10X? PacBio? Hi-C?), XX long read assemblies (if any not chromosome level? 10X? PacBio? Hi-C?), and XX short read (Illumina) assemblies."

-The figures fall considerably short of the standards I would anticipate for Nature Communications. I was also expecting a beefed-up supplement, with extended M&Ms and Results.

Minor edits; I advise the authors to proofread the complete ms.:

-We find evidence of extensive hybridization during the evolution of Xiphophorus based in both current and ancestral lineages." -> remove "based"

-"within the Xiphophorus genus" -> "within the genus Xiphophorus"

-"mitochondrial and nuclear nuclei" ???

-"Combining our robust species tree with the phylogeny of xmrk and egfrb supports a"... "Supplementary figure S6. Phylogeny tree of xmrk constructed.." -> This is a gene tree not a phylogeny!

-"Sword candidate genes. The sword candidate gene kcnh828" -> gene, not genes

Author Rebuttal letter:

Reviewer #1 (Remarks to the Author):

This study is a superb piece of work that leaves few stones unturned.

All elements of a good study are present:

The study system is interesting, complex, well-known yet still much to be discovered, with many important outstanding questions.

The sampling is excellent, all species are included

The approach is outstanding, full genomes are used

Thus, the setup is impeccable.

What I most appreciate is that studies that include a lot of genomes usually restrict their analyses to one topic: phylogeny, or one specific gene.

This study, instead, answers a lot of questions. The title, and the main goal, is to answer a fundamental evolutionary question, hybridization. This is a choice, and in my opinion a good one.

Meanwhile, a number of other very essential issues are dealt with as by catch from this study: melanoma (a topic that has been core to Xiphophorus research for decades) etc...

Overall, a huge pleasure to read this great paper.

Well written, nothing to change in my mind.

(and I don't think I have said that about reviewed papers in over a decade)

Response: We are very happy about this positive evaluation of our manuscript. This reviewer states that nothing should be changed. We took care that all changes that were made to the manuscript upon request of reviewer #2 did not alter what was highlighted by reviewer #1.

Reviewer #2 (Remarks to the Author):

-For a paper focused on phylogenomics, I find the phylogenomic analyses to be rather superficial. When examining the nDNA genomic dataset, differences arise between the concatenation ML and the Astral trees. However, it remains unclear what the primary driver of discordance is. Is it ILS? ("Discordance between gene trees and species trees can be generated by the coalescent process or by historical gene flow. We investigated this further using coalescent methods to build a nuclear phylogeny.") Authors should do more in depth analyses, looking at the sensitivity of the reconstructions. Additionally, since genomes are annotated, I anticipated seeing not only a protein-based tree but also trees based on CNNEs and UCEs (both concatenation ML and coalescent). Without more extensive analyses, it is challenging to identify the source of the incongruence.

Response: For the manuscript structure how to present our phylogenomics analyzes results, we 1,2 intended to maintain the research continuity from previous phylogenetic studies by emphasizing the mito-nuclear discordance. Hence discordance among the locus trees was not

presented as a focus. We regret if this has given the impression of "superficial analyses". Regarding nuclear phylogenomics, we have used whole genome DNA alignments, coding sequence alignments (partitioned), 4DTV sites alignments, and protein sequence alignments as signals to construct phylogenetic trees. In consideration of discordant evolutionary histories of genome mosaics, we have also constructed a coalescent phylogeny by defining recombination-free loci (1kb random windows that are 20kb away from each other) and synthesizing these locus-trees using ASTRAL.

Now following the reviewer's suggestion, we have further expanded the phylogenomic analyses by introducing new materials (conserved non-coding elements, CNEs) and new methods (CASTER3).

CNEs are similar to CNEEs and UCEs but offer more comprehensive information as the latter are retrieved by target-capture sequencing while the former is identified from whole genome sequencing. To identify CNEs, we first retrieved conserved regions from whole genome alignment using PhastCons4 and then removed the coding regions. With CNEs we have reconstructed the phylogeny using concatenation ML and coalescent method (Supplementary Fig. S12). The resulting topologies are almost the same as the previous results from whole genome alignments as phylogeny markers. The minor differences in the placement of *X. couchianus* and *X. meyeri* do not affect the conclusions of our paper.

We have also used CASTER to reconstruct a new coalescent tree from whole genome alignments. CASTER is a new product from the same lab that made ASTRAL and more recommended (<https://github.com/chaoszhang/ASTER>). The topology of the new coalescent tree is almost the same as the ASTRAL tree (Supplementary Fig. S11).

Differences arise between the concatenation ML and the ASTRAL trees, also between the mitochondrial and the nuclear trees. The incongruence may be caused by ILS or historical hybridization. To answer the question which is the primary driver we were trying to bring this up at the end of the section "Discordance of mitochondrial and nuclear phylogenetics", and to use it to introduce the next section "Extensive reticulate hybridization during the evolution of the genus *Xiphophorus*", in that section we attempt to distinguish hybridization from ILS by analyses such as AU-test and D-statistic/f₄-ratio based methods, and show that historical hybridization is the primary cause of incongruent phylogenies regardless of the effect of ILS. We thought that with this structure it is much easier for the non-expert to follow the logic of our argumentation. We have revised the manuscript to strengthen this logical flow and to avoid misunderstanding (line 240-244 and line 264-271).

Most importantly, given the pervasive introgression in the group (Fig. 2), why not run a set of analyses that exclude all introgressed loci? Such analyses might shed light on the sources of discordance regarding the unstable placement of several species, as well as the mitonuclear discordance reported (see below).

Response: We understand the reasoning behind the reviewer's comment here but believe that these analyses would not be methodologically rigorous for several reasons. First, analyses of introgression at a genome-wide scale highlight the proportion of the genome that has likely introgressed. However, for individual loci, it is difficult to distinguish between introgression and other causes of discordance, such as ILS or lack of sufficient phylogenetic information. Loci that can be more confidently identified as introgressed are likely to be a non-random subset of the data. For example, these loci may have more informative sites, have introgressed more recently, introgressed over greater phylogenetic distances, to name just a few possibilities. Thus, applying such filters to the dataset has the potential to introduce a large amount of systematic bias. As such, we respectfully disagree that this would be a reasonable change to our analysis.

The coalescent nDNA trees (and other trees that should be estimated as suggested above) should be included in Fig. 1 as well. A brief discussion about the incongruent results obtained with Astral concerning *X. clemenciae*, *X. monticolus*, and *X. mixei* should be added, as it renders the southern swordtail clade non-monophyletic. Is it an artifact arising from gene tree error, ILS, or is the introgression pulling clemen + mont closer to the platyfish? Additionally, the Astral tree is slightly less discordant with the mtDNA tree than the concatenation tree (perhaps again due to the introgression?).

Response: We agree that some arguments speak for including all those trees into Fig. 1. Accordingly, we have made such attempts. However, adding more information into this figure makes the reading rather difficult. Thus we added a new Fig. 2 to reveal the incongruence between concatenated- and coalescent-based trees. This leaves Fig. 1 focusing on the discordance of mitochondrial and nuclear phylogenies.

The CASTER tree and CNE-based trees are shown in the new Supplementary Fig. S11 and S12.

For the incongruent placement of the *X. clemenciae*/*X. monticolus*/*X. mixei* clade we have a short discussion in the result section: "The considerable amount of sequence originating

from outside of the Southern swordtails clade may explain the more basal position of the *X. clemenciae*/*X. monticulus*/*X. mixei* subclade in the coalescent tree. â

We agree with the reviewer that the ASTRAL tree is slightly less discordant with the mtDNA tree than the concatenation tree due to the incongruent placement of *X. clemenciae*/*X. monticulus*/*X. mixei*. This incongruent placement of ASTRAL tree does not resolve the mito-nuclear phylogeny discordance, and further indicates a hybridization happened in the ancestor lineage of this clade.

-I also wonder why no phylogenetic network analyses are presented, considering in particular that *Xiphophorus* is a model clade (if not the poster child) for reticulation. See <https://journals.plos.org/plosgenetics/article?id=10.1371/journal.pgen.1005896>. While I understand that phylogenetic networks can be computationally intensive to run, an alternative could be to use multiple independent subsets with fewer loci and fewer taxa.

Response: We have used Dsuite, a D statistics and f4-ratio based method, to identify reticulation in *Xiphophorus* from hybridization and introgression (see previous Fig. 2, now new Fig. 3A). We tried to run SNaQ in PhyloNet, but as the reviewer predicted, it was too computationally intensive to run for our complete dataset. Following the reviewer's suggestion, we have re-run this method with fewer taxa to confirm the reticulation happened in the *X. clemenciae*/*X. monticulus*/*mixei* clade (new Fig. 3B), and the *X. xiphidium* lineage (new Fig. 3C). The results are added and the paper is cited now (Fig. 3BC, line 281 and line 810-815).

-Did the authors include partitions in their phylogenetic analyses? It is less straightforward to do so with alignment windows/blocks that can include a plethora of locus type (exons, introns, flanking regions, UCEs, etc). But for protein coding genes, analyses should be partitioned minimally by codon position.

Response: We agree with the reviewer that it is less straightforward to include partitions with alignment windows. For this reason we did the partition with coding genes by codon position. However, we failed to explain this clearly in the method section. In the revised manuscript this is now described in detail (line 760).

-Two studies found a close association of *X. continens* with *X. pygmaeus*^{13,14} while all our trees place *X. continens* as sister taxon to *X. montezumae*. Our result is consistent with another recent study on Northern swordtails⁵⁵ Please comment on the source of discordance. Is it a miss ID in previous studies? (see also below).

Response: We comment on the source of discordance by saying: "Our whole genome re-sequencing of the material used in the conflicting studies revealed mistyping of the material"

-Genome annotation: you have a Table S6 reporting stats after annotations for all genomes, but it isn't intuitive for the reader to go there. Summarize the results from all those supplementary tables somewhere.

Response: We have added a remark about the content of Table S6. We are confident that readers will now understand the different values because we have added an explanation of the abbreviations in a footnote to the table.

I don't understand why simple things like building a mtDNA tree has so much detail, but other important approaches are barely explained in the main text (e.g., RNAseq).

Response: The mtDNA tree is a highly informative tool to investigate maternal lineages, which is important in our study to address the hybridization history in *Xiphophorus*. We want to emphasize for the reader that one way to identify hybridization is to reveal that the maternal and paternal origin were from distantly related species, which is obvious from the mito-nuclear phylogeny discordance. Not every reader will be familiar with the methods of retrieving mtDNA sequence from genome reads and building the trees. Hence we would like to give the information how this was done in our work.

We agree that RNAseq should be explained comprehensively if it was used for phylogeny construction or investigating gene functions. However, we only used RNAseq as one of the three evidence resources assisting to predict gene structures. In the method section, we have an explanation about how RNAseq is used for genome annotation. Nevertheless, following the reviewer's suggestion we have now added more information regarding the RNAseq in the Methods section (line 640-648).

-The resulting phylogenetic estimates were almost identical to the WGA tree

mtDNA tree nuDNA tree (Supplementary Fig. S8), except *X. xiphidium* and *X. alvarezii* 219 were placed in slightly different positions within their respective clade. *X. mayae* and *X. kallmani* also have unstable placements. Same goes for populations of *maculatus*. Please comment!
Response: The placement of *X. xiphidium* and *X. alvarezii* were consistent among the coding-region-based trees made from protein sequences, coding sequences and 4DTV sites repeatedly, but incongruent to the WGA tree. We point out in the revised manuscript that the incongruence may be caused by ILS or hybridization.
X. mayae and *X. maculatus*. Bp only show a different placement in the tree made from protein sequences, not that from CDS or 4DTV. We suspected an invalid analysis and thus did not mention it. The minor difference in placement does not affect the conclusion of our paper. *X. kallmani* is actually in a stable placement, it looks unstable only because of the inconsistent placement of *X. mayae*.

- Our whole genome resequencing of the material used in the conflicting studies revealed mistyping of the material (data not shown). Our analysis also strengthens the position of the Northern platyfishes species as being a crown group within the Southern platyfishes. Consequently, a distinction between both groups is no longer justified and we refer to the whole clade as "platyfishes". This statement has major taxonomic implications; the comparison with previous studies needs to be shown. Also, why choose "platyfishes" over "swordtails"? I think many more people are familiar with the latter common name (Google hits 725k for "platyfish" and 3.5M for "swordtail").
Response: The reviewer is correct that many more people may be more familiar with the term "swordtails" than with "platyfishes". However, both terms are the accepted common names for clades of the genus *Xiphophorus* and are used to identify the subgroups according to the NCBI taxonomy database and FishBase. Hence, with all respect we would like to keep both designations for distinguishing the subgroups.
For previous studies referring to the "Northern platyfishes" we have included a citation to a review on the taxonomy of fishes of the genus *Xiphophorus*.

- The bootstrap values of nodes were all at 100%. It is quite obvious that bootstrap support in WG phylogenomic analyses are inflated. Authors should use a more reliable metric of nodal support, like gene concordant factors and/or site concordant factors.
Response: We agree that it will make our inferences stronger by including the gene concordant factor (gCF) or site concordant factor (sCF) in our phylogenetic trees. This will reveal disagreement among locus- and site-phylogenies and hence serve as another evidence indicating the extensive hybridization history of the genus *Xiphophorus*. Given that we have already shown the locus based phylogeny discordance, we followed the reviewer's suggestion and calculated the sCF for both concatenated and coalescent trees using IQ-TREE2. Considering it would be too chaotic to put all the numbers onto the new Fig. 2 (the numbers and branch threads will block each other), we put the result into the new Supplementary Fig. S13.

- Trees with none of the nodes supported over 75% were removed, nodes with bootstrap value <75% were collapsed. The final coalescent tree was built based on the filter trees using ASTRAL-II. 75% seems like an overly astringent cutoff. The Astral tutorial recommends collapsing branches with less than 10% support:
<https://github.com/smirarab/ASTRAL/blob/master/astral-tutorial.md>. Try another cutoffs to see if the topology changes
Response: Following the reviewer's request, we have tried 10% as the cutoff for tree filtering. This resulted in exactly the same topology. We have added this information in the Materials and Methods section: "In an additional attempt, we used a supporting value of 10% as the cutoff for tree filtering before constructing the coalescent tree. This did not change the final topology."

- *xmrk* melanoma oncogene. The *xmrk* gene was originally found in the Southern platyfish, *X. maculatus*. From the whole genome information of all species, we were able to retrieve *xmrk* sequences in a total of nine platyfish and Northern swordtail species, but not in any of the Southern swordtails. Due to the high degree of conservation of *xmrk* and *egfrb* (its proto-oncogenic precursor) coding sequences, several nodes of the gene tree are only poorly supported (Supplementary Fig. S6). Despite this, a monophyletic origin of *xmrk* is visible. Given that most genomes were assembled using short reads, confirming absence would require

PCR. Minimally, refrain from making a strong case about the gene absence in some species.
Response: We agree that short read assemblies might miss genes, in particular local gene duplications. In consideration of this we had further confirmed the existence/absence of *xmrk* directly from the raw reads after screening the assembly sequences (see Method section "Retrieval of sequences for *xmrk* and *egfrb*"). Thus we are confident that the result for absence/presence is not affected by potential assembly failures.

Anyway, following the reviewer's suggestion, we have set up a PCR test. We designed primers which specifically amplify a region of *xmrk* and the homologous region from *egfrb*. Using a restriction enzyme that cuts the PCR product of *egfrb* but not *xmrk*, we confirmed the presence of *xmrk* by observing PCR products resistant to the enzyme. The absence of *xmrk* was confirmed by observing all PCR products cleaved into two fragments of the expected size for *egfrb* (new Supplementary Fig. S8).

- From the genome assemblies we were able to identify multiple mutant copies, which are predicted to be defective, also in *X. maculatus*, *X. xiphidium*, *X. birchmanni*, *X. malinche*, *X. evelynae* and *X. sp III*, but not in any of the Southern swordtails. Genome assemblies can be compromised by sequencing error (in particular short-read genomes). Again, ruling out that option requires PCR.

Response: Due to the many different ways how mutations affected the *mc4r* ORF (5' and 3' truncations, large and small indels, point mutations, frameshifts) we cannot design primers that would be suited for a PCR test for absence of defective copies. We agree that short read gene assemblies might be compromised. To make clear that we cannot rule out completely the presence of defective alleles we have included a note of caution here: "Because genome assemblies can be compromised by sequencing errors (in particular short-read genomes) further analyses are necessary to confirm absence of such defective copies."

- To collect transcriptome gene evidence, RNA-seq reads from multiple tissues were cleaned using fastp96, and were mapped onto the assembly using HISAT2 -> which species
Response: This is a general description of the genome annotation pipeline that was used for all species in the study.

- The size of certain gene families also varies significantly across the genus 403 *Xiphophorus* (Supplementary Table S7). While these patterns have been previously 404 reported for *paox* and *mc4r46*, we identify exciting new patterns of gene family expansion 405 in our genome-wide dataset. Perhaps most notable given the importance of sexual selection 406 in *Xiphophorus* is the identification of diversification in gene families associated with vision and olfactory functions. There's no way for the reader to see this. Table S7 contains all the raw data output from the CAFE analysis. This needs to be summarized somehow for ease of interpretation. Also, olfactory gene families are always expanded - that's not something unique to swordfishes.

Response: We agree with the reviewer that table S7 contains too many details which may dazzle readers. However, this might be inevitable since, instead of focusing on a single lineage like in other similar analyses, we are trying to reveal the family size dynamic for a whole genus. We have a summary for CAFE analysis in Supplementary Fig. S4 but in that case gene family names can not be shown. To alleviate this problem, we have colored in table S7 those gene families that were commented in the manuscript. In addition, we made a heatmap to visualize the result (Supplementary Fig. S5).

The reviewer is correct. The dynamic of the olfactory gene family, expanded in some categories while contracted for some others, is a common and flexible approach for life to be adaptive, not unique to swordfishes. However it is interesting and novel to reveal the olfactory gene dynamic in the micro-evolution of a genus, where closely related species possess different categories and amounts of olfactory genes as a potential genetic basis for niche diversity and reproductive isolation.

- Here, we add twenty-nine additional genomes to these existing genomic resources, which were sequenced using Illumina, 10X, PacBio and/or Hi-C techniques in this study (Supplementary Table S1). This now provides a complete genome resource of all *Xiphophorus* species, including all 26 previously identified species (<https://www.ncbi.nlm.nih.gov/Taxonomy/Browser/wwwtax.cgi?id=8082>), three undescribed taxa, *X. sp I*, *X. sp II*, *X. sp III*, two new strains of *X. maculatus*, and a reference genome for the species *Priapella lacandonae* as outgroup.

This is misleading. Most genomes were sequenced using Illumina short reads (shotgun genomes). As written, the ms. gives the impression that most new genomes are chromosome level. Clarify! State for example, "we add twenty-nine additional genomes to these existing genomic resources, including XX chromosome level genomes (10X? PacBio? Hi-C?), XX long read assemblies (if any not chromosome level? 10X? PacBio? Hi-C?), and XX short read (Illumina) assemblies."

Response: This is now clarified. In addition, we pointed out those chromosome-level assemblies in the last paragraph of this section: "Those genomes, which are assembled at chromosome level (*X. birchmanni*, *X. malinche*, *X. couchianus*, *X. maculatus*, *X. hellerii* and *P. lacandona*)...". When presenting the assembly results, we were focusing on what species we have provided the genome resource.

-The figures fall considerably short of the standards I would anticipate for Nature Communications. I was also expecting a beefed-up supplement, with extended M&Ms and Results.

Response: We have improved the figures by tuning the color scheme and adding more information. We are open for further suggestions or instructions by the Nature Communications editorial team.

For our work we were able to include all results and methods in the main manuscript. We made specific efforts to make our Materials and Methods section fully informative and thus did not need to write further details in a separate file. We have 15 Supplementary Figures and 7 Supplementary Tables. If this reviewer is missing important information we are happy to include this upon specification.

Minor edits; I advise the authors to proofread the complete ms.:

Response: We have proofread the complete ms. and corrected typos and grammatical errors.

-We find evidence of extensive hybridization during the evolution of *Xiphophorus* based in both current and ancestral lineages." -> remove "based"

Response: Done.

-"within the *Xiphophorus* genus" -> "within the genus *Xiphophorus*"

Response: Done.

-"mitochondrial and nuclear nuclei" ???

Response: Changed to "mitochondrial and nuclear sequences".

-"Combining our robust species tree with the phylogeny of *xmrk* and *egfrb* supports a Supplementary figure S6. Phylogeny tree of *xmrk* constructed.. -> This is a gene tree not a phylogeny!

Response: Corrected.

-"Sword candidate genes. The sword candidate gene *kcnh828*" -> gene, not genes

Response: This paragraph reports our results also for the second sword candidate gene *sp8*.

Therefore we think that the plural "genes" in the heading of this paragraph is correct.

1 Kang, J. H., Scharl, M., Walter, R. B. & Meyer, A. Comprehensive phylogenetic analysis of all species of swordtails and platies (Pisces: Genus *Xiphophorus*) uncovers a hybrid origin of a swordtail fish, *Xiphophorus monticolus*, and demonstrates that the sexually selected sword originated in the ancestral lineage of the genus, but was lost again secondarily. *BMC evolutionary biology* 13, 1-19 (2013).

2 Cui, R. et al. Phylogenomics reveals extensive reticulate evolution in *Xiphophorus* fishes. *Evolution* 67, 2166-2179 (2013).

3 Zhang, C., Nielsen, R. & Mirarab, S. CASTER: Direct species tree inference from whole-genome alignments. *bioRxiv*, 2023.2010.2004.560884, doi:10.1101/2023.10.04.560884 (2023).

4 Siepel, A. et al. Evolutionarily conserved elements in vertebrate, insect, worm, and yeast genomes. *Genome Research* 15, 1034-1050, doi:10.1101/gr.3715005 (2005).

Version 1:

Reviewer comments:

Reviewer #2

(Remarks to the Author)

I commend the authors for addressing my previous point. I have no further comments.
